# Ube2D3 and Ube2N are essential for RIG-I-mediated MAVS aggregation in antiviral innate immunity

Yuheng Shi[1,*], Bofeng Yuan[1,*], Wenting Zhu[1,*], Rui Zhang[1], Lin Li[2], Xiaojing Hao[1], She Chen[2] & Fajian Hou[1]

Innate immunity plays a pivotal role in virus infection. RIG-I senses viral RNA and initiates an effective innate immune response for type I interferon production. To transduce RIG-I-mediated antiviral signalling, a mitochondrial protein MAVS forms prion-like aggregates to activate downstream kinases and transcription factors. However, the activation mechanism of RIG-I is incompletely understood. Here we identify two ubiquitin enzymes Ube2D3 and Ube2N through chromatographic purification as activators for RIG-I on virus infection. We show that together with ubiquitin ligase Riplet, Ube2D3 promotes covalent conjugation of polyubiquitin chains to RIG-I, while Ube2N preferentially facilitates production of unanchored polyubiquitin chains. In the presence of these polyubiquitin chains, RIG-I induces MAVS aggregation directly on the mitochondria. Our data thus reveal two essential polyubiquitin-mediated mechanisms underlying the activation of RIG-I and MAVS for triggering innate immune signalling in response to viral infection in cells.

[1] State Key Laboratory of Cell Biology, CAS Center for Excellence in Molecular Cell Science, Innovation Center for Cell Signaling Network, Shanghai Institute of Biochemistry and Cell Biology, Chinese Academy of Sciences, University of Chinese Academy of Sciences, 320 Yueyang Road, Shanghai 200031, China. [2] National Institute of Biological Sciences, Beijing 102206, China. * These authors contributed equally to this work. Correspondence and requests for materials should be addressed to F.H. (email: fhou@sibcb.ac.cn).

Viral infection evokes both innate and adaptive immune responses in higher organisms[1]. As the first line of defense, an antiviral innate immune response usually results in the rapid production of type I interferon (IFN) and proinflammatory cytokines and therefore induces an antiviral state[2,3]. Antiviral innate immune response is triggered by the ligation of pathogen associated molecular patterns (PAMP) and pattern recognition receptors (PRR)[4]. RIG-I-like receptors (RLR) are major innate immune sensors for viral RNA[5]. Viral RNA activates RIG-I, which transduces antiviral signal through the adaptor protein MAVS, also known as VISA, IPS-1 and Cardif[6–9]. MAVS activates downstream effectors such as TBK1 and IKK. TBK1 and IKK in turn activate transcription factors IRF3 and NF-κB and promote their translocations to the nucleus to turn on the expression of type I IFN and other cytokines.

Structural studies have shown that RIG-I binds to viral RNA through its central helicase domain and carboxyl (C)-terminal domain (CTD), which then releases its amino (N)-terminal tandem caspase activation and recruitment domains (2CARD) for homotypic interaction with MAVS N-terminal CARD domain[10–13]. On interaction with RIG-I 2CARD, MAVS forms prion-like filament and releases its active regions to recruit downstream signalling molecules[14,15]. Notably, MAVS filament formation is a hallmark of its activation and essential for its antiviral function[15,16], highlighting the importance of the molecular mechanism underlying MAVS aggregation induced by RIG-I. Ubiquitination of RIG-I was initially shown to be a critical element for it to activate MAVS in cells[17]. However, RIG-I was also shown to form filament along double-stranded RNA (dsRNA) in a ubiquitin-independent manner, which promotes its 2CARD oligomerization and sufficiently stimulates MAVS in vitro[18].

Ubiquitination is an important posttranslational modification for proteins involved in a variety of biological processes by altering their stability, localization or interaction property. Ubiquitin modification to a target protein is accomplished through an enzymatic cascade involving three enzymes, ubiquitin-activating enzyme (E1), ubiquitin-conjugating enzyme (E2) and ubiquitin ligase (E3)[19]. A ubiquitin moiety is usually attached to specific lysine residues of a protein substrate by its C-terminal glycine residue (G76), and polyubiquitin chains are usually formed by sequential attachment of the next ubiquitin molecule to one of its seven internal lysine residues or N-terminal methionine residue, resulting in different polyubiquitin chain linkages (K6, K11, K27, K29, K33, K48, K63 and linear). There are 2 E1s, about 50 E2s and 700 E3s in human genome[20]. E2–E3 pairs determine the specificity of polyubiquitin chain linkage and protein substrate. TRIM25 was initially identified as an essential E3 ubiquitin ligase to interact with RIG-I 2CARD and promote its ubiquitination at lysine172 (ref. 17). Subsequently, other E3 ubiquitin ligases such as Riplet (also known as RNF135/REUL), MEX3C and TRIM4, were reported to regulate RIG-I pathway positively, through a covalent conjugation of polyubiquitin chains to RIG-I[21–24]. However, both the relative contributions of these E3 ubiquitin ligases to RIG-I activation and the coordination of their functions in RIG-I-mediated antiviral signalling are poorly understood. There is no direct evidence showing the modification sites of full-length RIG-I by ubiquitin moiety. On the other hand, unanchored polyubiquitin chains were shown to be responsible for RIG-I to activate MAVS as an alternative mechanism[25]. In support of this model, unanchored polyubiquitin chains were shown to induce tetramerization of RIG-I 2CARD, which was competent in inducing MAVS aggregation and activation in vitro[26,27]. Nevertheless, the origin of unanchored polyubiquitin chains is unknown. Collectively, the mechanisms of ubiquitin-mediated RIG-I activation remain obscure.

In this report we reconstitute a cell-free assay as a direct measurement for viral RNA-triggered activation of RIG-I and MAVS. Following this assay, we study essential factors required for RIG-I to induce MAVS aggregation and activation. With loss-of-function analysis in both human and mouse cells, we determine that Riplet, rather than other E3 ubiquitin ligases, such as TRIM25 as previously reported, is the only E3 ubiquitin ligase that is required for RIG-I to activate MAVS in the early phase production of type I IFN. Through biochemical purification, we identify two ubiquitin-conjugating enzymes, Ube2D3 (also known as Ubc5c) and Ube2N (also known as Ubc13), as essential E2s in RIG-I activation. Furthermore, we show that Ube2D3-Riplet pair was able to conjugate polyubiquitin chains covalently to RIG-I to trigger MAVS aggregation. In contrast, the Ube2N-Riplet pair could catalyse the formation of unanchored polyubiquitin chains, which is also potent in facilitating MAVS aggregation and activation by RIG-I.

## Results

**A cell-free assay of RIG-I-mediated MAVS aggregation.** To dissect the biochemical mechanism underlying RIG-I and MAVS activation in antiviral signalling, we developed a cell-free assay using MAVS aggregation as the readout. HEK293T cells were homogenized to obtain crude cellular lysate. To mimic viral infection-induced innate immune signalling in cells, the genomic RNA was extracted from vesicular stomatitis virus (VSV) and used as a stimulus. Addition of VSV RNA (vRNA) to HEK293T crude lysate triggered MAVS aggregation efficiently, as shown by semi denaturing detergent agarose gel electrophoresis (SDD–AGE) (Fig. 1a). Meanwhile, IRF3 dimerization was detected in the reaction mixture, suggesting that the observed MAVS aggregates are able to transduce signal to downstream effectors. In the cell-free assay, MAVS aggregation was dependent on RIG-I, as vRNA could not induce MAVS aggregation in $Rig\text{-}i^{-/-}$ cell lysate under the same condition. Strikingly, adding back full-length RIG-I recombinant protein to the cell-free assay could restore MAVS aggregation induced by vRNA. These data suggested that the cell-free assay faithfully recapitulates RIG-I-mediated MAVS aggregation in antiviral innate immune signalling (Supplementary Fig. 1a). Notably, RIG-I-2CARD could trigger MAVS aggregation in a manner independent of endogenous RIG-I or vRNA, indicating a different mechanism from the activation of endogenous RIG-I. Therefore, we used full-length RIG-I recombinant protein in the cell-free assay to investigate relevant biochemical mechanisms in the following study.

**Ubiquitination is required for MAVS aggregation.** To determine if polyubiquitin chains are involved in MAVS activation as reported previously, we utilized two chemicals to prevent ubiquitination in the cell-free assay. Before addition of vRNA, the reaction mixture was pretreated with N-ethylmaleimide (NEM), an organic compound that inactivates E1 and E2 by modifying their active site cysteine residues, or PYR-41, which disables E1 specifically. As shown in Fig. 1b, NEM or PYR-41 treatment blocked MAVS aggregation in the presence of vRNA, suggesting that an ubiquitination event is required for MAVS aggregation following vRNA addition. As expected, PYR-41 treatment inhibited MAVS aggregation in HEK293T cells on virus infection, indicating that a ubiquitination event is indeed critical for RIG-I to activate MAVS following virus infection (Fig. 1c).

A panel of E3 ubiquitin ligases, including TRIM25, Riplet, TRIM4 and MEX3C, were reported to be involved in RIG-I-mediated antiviral signalling, which are presumably required for MAVS aggregation and activation. To find out the

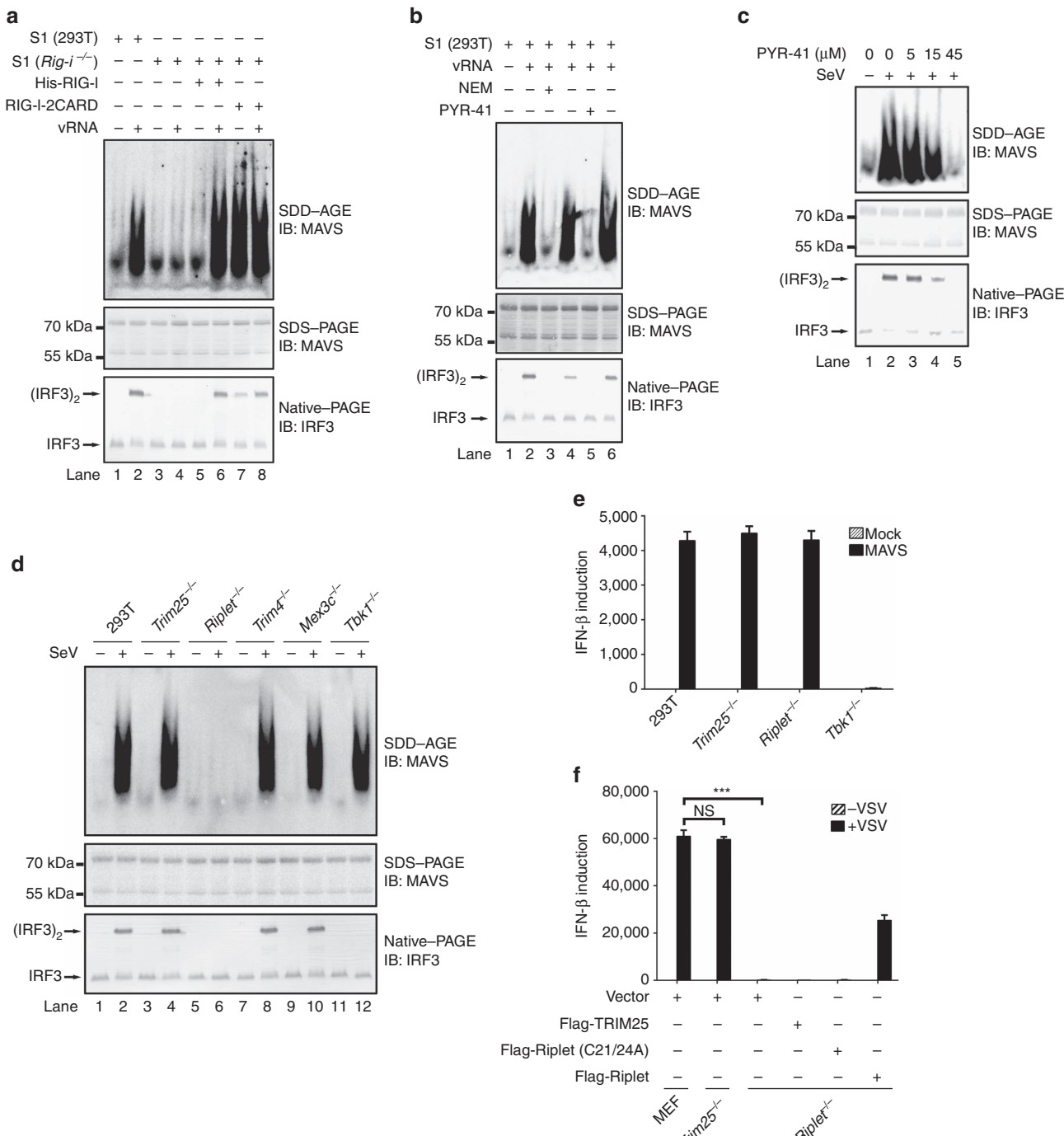

**Figure 1 | Reconstitution of a cell-free assay recapitulating RIG-I and MAVS activation in antiviral signalling.** (**a,b**) S1 fraction from HEK293T cells (wild type or *Rig-i*−/−) was incubated with or without His-RIG-I, His-RIG-I-2CARD, vRNA at 30 °C for 1 h, and NEM (40 μM) or PYR-41 (40 μM) was included as indicated (**b**). The reaction mixture was then separated with SDD–AGE, SDS-PAGE and Native-PAGE followed by immunoblotting (IB) analysis. The original full blot for SDD–AGE shown in **a** can be found in Supplementary Fig. 8a. (**c**) HEK293T cells were treated with PYR-41 at indicated concentrations for two hours before infected with or without Sendai virus, which were collected for subcellular fractionation followed by immunoblotting. (**d**) HEK293T cells (wild type and knockout lines) were infected with or without Sendai virus, which were collected 12 h post infection and analysed as described in **d**. See also Supplementary Fig. 1b. The original full blot for SDD–AGE can be found in Supplementary Fig. 8b. (**e**) HEK293T cells (wild type and knockout lines) were transfected with pcDNA3-flag-MAVS. Thirty six hours after transfection, the cells were collected and IFN induction was measured with quantitative PCR (qPCR). See also Supplementary Fig. 1e. (**f**) Flag-Riplet (wild type or mutant form) and Flag-TRIM25 were transduced into MEFs (wild type and knockout lines) by retrovirus. Forty eight hours after transduction, the cells were infected with VSV for sixteen hours. The cells were then collected and IFN induction was measured with qPCR. See also Supplementary Fig. 1f. *P<0.05 and ***P<0.001. NS indicates no statistically significant difference. See also Supplementary Fig. 1g.

essential E3 ligase(s) involved in MAVS activation, we made HEK293T cell lines with these genes knocked-out individually and examined their effects on MAVS aggregation. Surprisingly, MAVS aggregation was abrogated only in $Riplet^{-/-}$ cells following Sendai virus infection, but not in other knockout cell lines analysed (Fig. 1d). Consistently, type I IFN production correlated with MAVS aggregation in these knockout cell lines on virus infection (Supplementary Fig. 1c). The induction of a number of cytokines was also abolished in $Riplet^{-/-}$ but not $Trim25^{-/-}$ or $Mex3c^{-/-}$ cells on virus infection. In addition, exogenous expression of MAVS in $Riplet^{-/-}$ cells could induce IFN production, suggesting that Riplet is not required for antiviral signalling downstream of MAVS (Fig. 1e). Interestingly, overexpression of RIG-I-2CARD could also induce IFN production in $Riplet^{-/-}$ cells, suggesting ectopically expressed RIG-I-2CARD could be activated by polyubiquitin chains synthesized by other E3s (Supplementary Fig. 1d).

To further validate these findings, we investigated RIG-I signalling in mouse embryonic fibroblast (MEF). Strikingly, knockout of $Trim25$ had marginal effect on IFN production in MEFs in response to virus infection, while knockout of $Riplet$ abolished IFN production completely (Fig. 1f). The defect of IFN production in $Riplet^{-/-}$ MEFs could be rescued by exogenous expression of wild type Riplet but not the mutant form that harbours two C-to-A mutations at the active site cysteine residues 21 and 24 in its RING domain. In addition, crude cell lysate from these cell lines were tested in the cell-free assay. Consistently, vRNA could trigger MAVS aggregation in wild type and $Trim25^{-/-}$ cell lysate, but not in $Riplet^{-/-}$ cell lysate (Fig. 2a). Supplement of Riplet recombinant protein to $Riplet^{-/-}$ cell lysate restored MAVS aggregation in response to vRNA, while the mutant form of Riplet failed to rescue. Taken together, our data suggest that Riplet but not TRIM25 is essential for RIG-I to induce MAVS aggregation in antiviral signalling both in cells and in the cell-free assay.

**Identification of ubiquitin E2 enzymes for MAVS aggregation.** Having established the pivotal role of Riplet in RIG-I and MAVS aggregation, we aimed to identify unknown factors involved in the process. To facilitate the investigation on unknown factors, known factors in the cell-free assay were expressed and purified as recombinant proteins to substitute their endogenous counterparts (Supplementary Fig. 2a). Thereafter, the assay was refined to include ubiquitin, E1, Riplet, RIG-I, S100 (containing cytosolic proteins) and P5 fraction (containing mitochondria and MAVS; Fig. 2b). S100 is indispensable for MAVS aggregation in the refined cell-free assay, indicating S100 contains an unknown factor(s) required for RIG-I activation. S100 was first separated into three fractions (A, B and C) by a HiTRAP Q-sepharose anion exchange column chromatograph. Fraction A flowed through the Q column without binding to it. Combination of fractions A and B were able to substitute S100 to induce MAVS aggregation, which was called active fraction (Fig. 2c). We further found that fraction B or C failed to trigger MAVS aggregation but were instead found to contain known factors such as E1 (Supplementary Fig. 2c). These results indicated that Fraction A might contain unknown essential factor(s) for MAVS aggregation.

We then devised a strategy to further purify the active component(s) in Fraction A by sequential chromatographs over an array of columns (Supplementary Fig. 2d). At the last step, we performed Superdex-200 size exclusion chromatograph and collected fractions to test their ability to induce MAVS aggregation. We found four active fractions, that is, Fraction 4–7, each of which was able to substitute S100 to induce MAVS aggregation in the cell-free assay (Fig. 2d, middle panel). After

SDS–PAGE, one common band of ∼15 kDa present in all four active fractions was visualized by silver staining (Fig. 2d, top panel). This band was excised and subjected to protein identification. Peptides identified by mass spectrometric analysis pinpointed several E2 ubiquitin-conjugating enzymes, including Ube2D2, Ube2L3, Ube2D3, Ube2D4, Ube2D1, Ube2E1, Ube2N and Ube2M (Supplementary Table 1). As expected, immunoblotting confirmed the presence of these E2s in all four active fractions from size exclusion chromatograph (Supplementary Fig. 2e). Among these E2s, Ube2D1, 2, 3 and 4 (also known as UbcH5a, b, c and d) show a high degree of sequence homology and belong to one family (collectively called Ube2Ds). Ube2N (also known as Ubc13) forms a heterodimeric complex with Ube2V1 (Uev1A) to specifically catalyse K63-linked polyubiquitin chain formation.

**Ube2D3 and Ube2N are required for RIG-I and MAVS activation.** E2s identified above were expressed and purified as recombinant proteins to be used in the cell-free assay (Supplementary Fig. 3a). Strikingly, Ube2Ds, Ube2E1 and Ube2N robustly triggered MAVS aggregation in the presence of vRNA (Fig. 3a,b). This effect of the E2s are dependent on their catalytic activity, as their respective catalytically dead mutants, in which the active site cysteine residues were disrupted, failed to support MAVS aggregation in the cell-free assay (Supplementary Fig. 3b).

We next explored whether these E2s are indeed required for RIG-I and MAVS activation in cells on virus infection. We took a loss-of-function approach beginning with knocking-out genes encoding these E2s in HEK293T cells. Deficiency in any individual one of these genes showed marginal effect on MAVS aggregation on virus infection, suggesting they might play redundant roles (Supplementary Fig. 3c). We then tried to knockout multiple E2 genes in one cell cline. We obtained cell lines with $Ube2D1/2/4^{-/-}$, $Ube2D1/2/4$ and $Ube2N$ and $Ube2E1^{-/-}$ successfully, but we could not attain to lines with $Ube2D1/2/3/4^{-/-}$ or $Ube2D3$ and $Ube2N^{-/-}$. One explanation is that $Ube2D1/2/3/4^{-/-}$ or $Ube2D3$ and $Ube2N^{-/-}$ are lethal to the cell, which is consistent with a previous report and the personal communication with Dr Galanty[28]. Therefore, we decided to knock-down $Ube2D3$ expression with small interfering RNA (si-RNA) in the context of knockout cell lines obtained. As shown in Fig. 3c, disruption of both $Ube2D3$ and $Ube2N$ expression abrogated the IFN induction following virus infection. We noticed that on viral infection, $Ube2D1/2/4^{-/-}$ and $Ube2N/E1^{-/-}$ cells produced more IFN than $Ube2N^{-/-}$ cells, which could be due to the negative effect of those E2s on IFN-β (Supplementary Fig. 3d). Consistently, MAVS could not form aggregates on virus infection in the absence of both Ube2D3 and Ube2N (Fig. 3d). These results strongly suggested that $Ube2D3$ and $Ube2N$ play redundant roles in MAVS aggregation, while $Ube2E1$ and $Ube2D1/2/4$ might be dispensable. In support of this conclusion, knocking-down of Ube2D3 did not affect the expression of Ube2D1/2/4 (Fig. 3e). Furthermore, to exclude the possibility of any off-target effect incurred during gene editing, we tried to rescue the loss-of-function phenotype with exogenous expression of Ube2D3 or Ube2N. Remarkably, when Ube2Ds or Ube2N was ectopically expressed in si-Ube2D3/$Ube2N^{-/-}$ cells, MAVS aggregation could be fully restored on virus infection (Fig. 3f), and IFN induction was also rescued (Supplementary Fig. 3e). Altogether, these data suggested that Ube2Ds and Ube2N play redundant and critical roles in MAVS aggregation in antiviral innate immune signalling.

**Ube2N is essential for MAVS aggregation in MEF/PEM/BMDM.** We also examined the involvement of Ube2D3 and Ube2N in

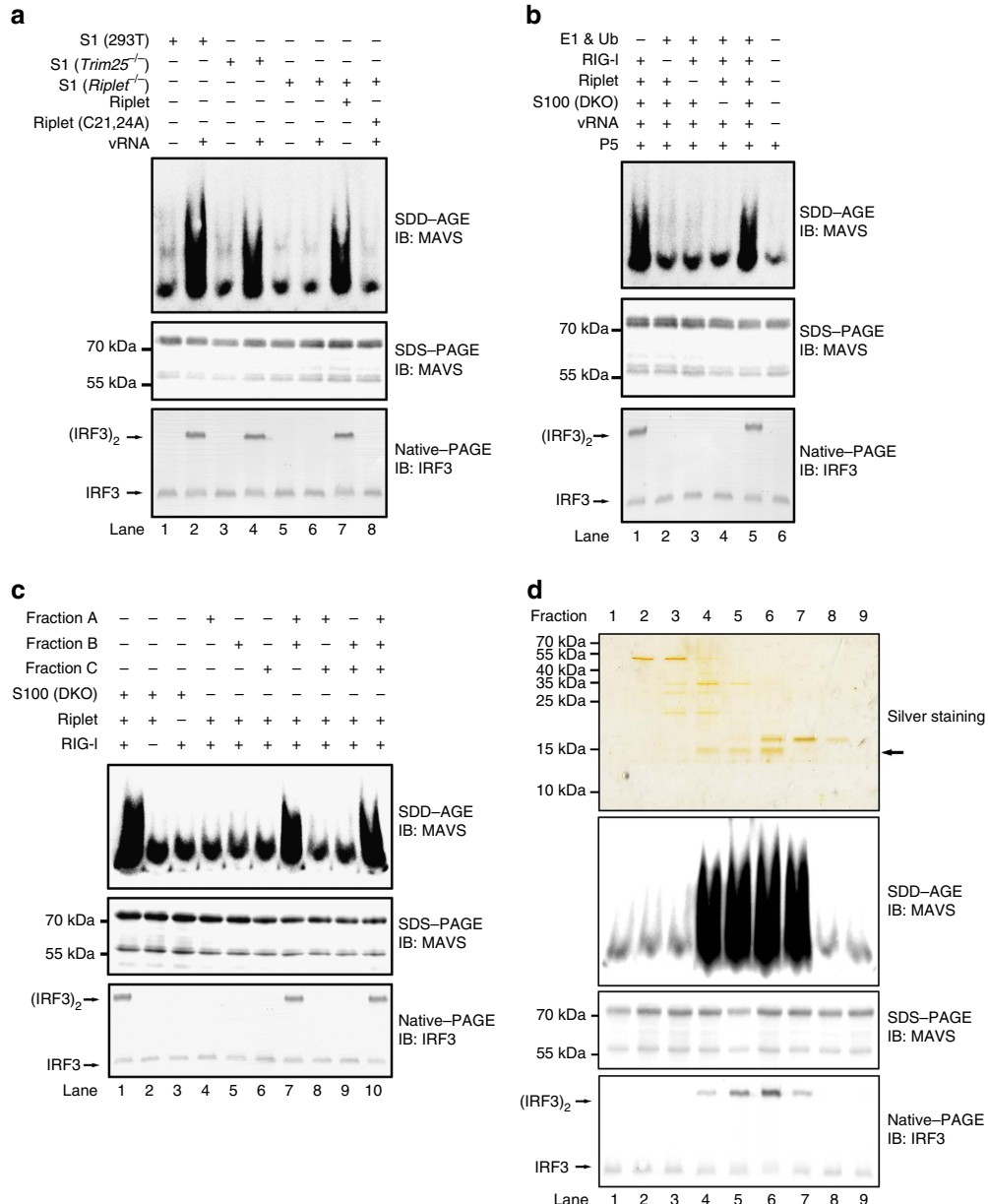

**Figure 2 | Purification of multiple E2s required for RIG-I and MAVS activation in the cell-free assay.** (**a**) The cell-free assay was performed in the presence or absence of Riplet as described in Fig. 1a. (**b**) S100 fractions were prepared from HEK293T cells (*Rig-i*$^{-/-}$ and *Riplet*$^{-/-}$, DKO). The cell-free assay containing S100 and vRNA was performed with or without purified components (E1, Ub, RIG-I and Riplet) as indicated. One hour after incubation, the reaction mixture was further incubated with P5 fraction. P5 was then subjected to SDD-AGE and IRF3 dimerization assay. See also Supplementary Fig. 2b. (**c**) Fractions (A, B and C) from Q column were used to replace S100 fraction in the cell-free assay and analysed as described in (**b**, Lane 5). (**d**) Fractions (from 1 to 9) from Superdex200 column were used to replace S100 fraction in the cell-free assay and analysed as described in **b**. These fractions were also separated by SDS-PAGE and visualized by silver staining (top panel). The band indicated by the arrow were isolated and subjected to mass spectrometric analysis. The original full blot for silver-staining can be found in Supplementary Fig. 8c.

antiviral innate immune signalling in MEFs. Small hairpin RNAs (shRNA) targeting these E2s were introduced into MEFs to abate their expression, respectively. Surprisingly, sh-Ube2N alone severely suppressed IFN (α and β) induction in response to virus infection (Fig. 4a,b), while sh-Ube2Ds decreased IFN induction slightly. We noticed that sh-Ube2Ds treatment inhibited Ube2N expression somehow, which could lead to the decreased IFN induction. Similarly, the expression of other cytokines (CXCL10 and IL6) was also markedly crippled in the absence of Ube2N (Supplementary Fig. 4a,b). Immunoblotting showed that sh-Ube2N did not affect the expression level of Ube2Ds (Fig. 4c). Meanwhile, ectopically expressed MAVS in sh-Ube2N-treated

MEFs induced IFN (Fig. 4d) and IL6 (Supplementary Fig. 4c) potently, suggesting that Ube2N might not be important for MAVS downstream signalling and is instead critical for MAVS activation in MEFs. Indeed, MAVS could not form prion-like aggregates under sh-Ube2N treatment in response to virus infection (Fig. 4e). Consistently, P5 fraction isolated from sh-Ube2N-treated MEFs showed no activity in stimulating IRF3 dimerization, suggesting that MAVS was not activated on virus infection. Notably, the defect of IFN induction in the absence of Ube2N could be rescued by ectopically expressed Ube2N, ruling out the possibility of off-target effect (Fig. 4f). As expected, Ube2D3 was not able to rescue the defect of IFN induction by

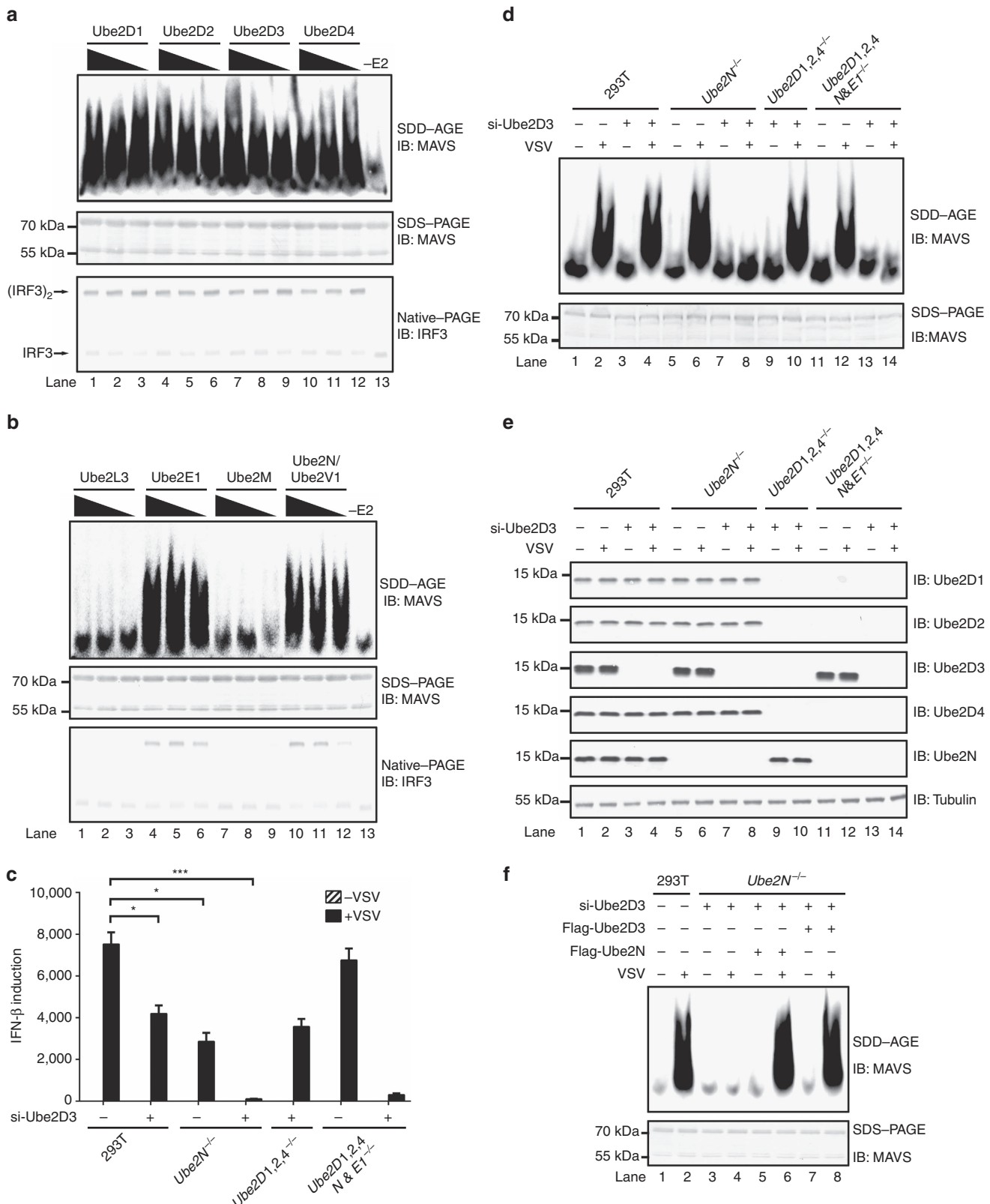

viral infection in sh-Ube2N-treated cells. Consistently, when Ube2N was knocked down, IFN production in response to viral infection was severely crippled in other mouse primary cells, such as peritoneal macrophage (PEM) and bone marrow-derived macrophage (BMDM; Supplementary Fig. 5a,b). Knocking-down

experiments also confirmed the critical role of Riplet but not TRIM25 in IFN production in these cells (Supplementary Fig. 5c–f). Taken together, these results suggest that Ube2N play an essential role for RIG-I-mediated MAVS activation in MEFs.

**Associations of RIG-I with Riplet and Ube2D3/Ube2N**. Having established the essential roles of Ube2D3 and Ube2N in MAVS aggregation triggered by RIG-I, we next investigated how these essential components are recruited to accomplish an effective ubiquitination. Epitope-tagged RIG-I and Riplet were coexpressed in HEK293T cells. Immunoprecipitation (IP) of RIG-I pulled-down significant amount of Riplet, indicating they interact with each other, and this interaction was enhanced on virus infection (Fig. 5a, Supplementary Fig. 6a). Deletion of RIG-I CARD domain but not its RD domain severely impaired the interaction between RIG-I and Riplet (Fig. 5b, Supplementary Fig. 6b). Importantly, RIG-I 2CARD alone was able to interact with Riplet, suggesting that Riplet may facilitate the modification of RIG-I 2CARD domain (Fig. 5c). Further analysis by reciprocal IP revealed that RIG-I bound to Riplet through its SPRY domain (Fig. 5d). As expected, our results showed that Riplet binds to Ube2D3 and Ube2N, with preference to the latter (Fig. 5e). Interestingly, RIG-I was able to interact with both Ube2D3 and Ube2N (Fig. 5f).

**Two mechanisms underlying RIG-I and MAVS activation**. Using purified components (that is, RIG-I, Riplet, Ube2D3 or Ube2N, E1 and ubiquitin) in the cell-free assay, we analysed the biochemical mechanism underlying RIG-I and MAVS activation by ubiquitination. After incubation of RIG-I with ubiquitin enzymes (E1, E2 and E3), the reaction mixture was treated with deubiquitination enzymes (DUB) before further incubation with P5 fraction to examine MAVS aggregation. One of the DUBs, the viral OTU (vOTU), destroys most of polyubiquitin chains, including unanchored ones and those that are covalently conjugated to a target protein. The other DUB, IsoT, specifically cleaves unanchored polyubiquitin chains but not those that are covalently conjugated to a target protein[25,29]. As expected, vOTU abolished both Ube2N- and Ube2D3-mediated MAVS aggregation (Fig. 6a). Interestingly, IsoT abrogated Ube2N- but not Ube2D3-mediated MAVS aggregation. These results suggested that Ube2N facilitates the synthesis of unanchored polyubiquitin chains for RIG-I and MAVS activation, while Ube2D3 promotes the covalent conjugation of polyubiquitin chains to RIG-I. Indeed, shifting of RIG-I on SDS–PAGE could be readily detected after incubation with Ube2D3 and Riplet, suggesting that RIG-I was covalently modified by polyubiquitin chains. In contrast, no shifting of RIG-I could be detected after incubation with Ube2N and Riplet, indicating that unanchored polyubiquitin chains were synthesized, which were not conjugated to RIG-I.

To further validate our finding that Ube2N mediates the production of unanchored polyubiquitin chains for RIG-I activation, we performed the ubiquitination reaction without RIG-I. Ubiquitin enzymes were incubated together with ubiquitin in the absence of RIG-I, and NEM was then added to stop the reaction. Strikingly, the reaction mixture showed a potent activity in inducing MAVS aggregation and activation when RIG-I was

supplemented (Fig. 6b, Lane 4), which could be abolished by IsoT treatment (Fig. 6b, Lane 3). This result demonstrated that Ube2N-Riplet catalyses the synthesis of unanchored polyubiquitin chains. Furthermore, we compared the activity of Ube2D3-Riplet pair with Ube2D3-TRIM25 pair in our cell-free assay. Consistent with a previous report, our result showed that TRIM25 and Ube2D3 promotes the synthesis of unanchored polyubiquitin chains, which could induces MAVS aggregation in the presence of RIG-I and was sensitive to IsoT treatment (Fig. 6c, Lane 4)[25]. In contrast, Riplet and Ube2D3 could catalyse the covalent conjugation of polyubiquitin chains to RIG-I for its activation, which was resistant to IsoT treatment (Fig. 6c, Lane 3).

To determine the polyubiquitin chain linkage involved, we tested the ability of various forms of ubiquitin in promoting RIG-I activation in the cell-free assay. Single point mutation K63R, which disrupted only lysine63 residue, abolished RIG-I activation completely, while another mutant K63, in which all lysine residues except lysine63 were mutated, could fully support RIG-I activation (Fig. 6d). Similar results were obtained when Ube2N was used as an E2 (Fig. 6e). K63-linkage-specific antibody against ubiquitin chains revealed that Ube2N-Riplet synthesized K63-linked polyubiquitin chains specifically (Supplementary Fig. 6c). Collectively, our data suggested that K63-linked polyubiquitin chains are essential in both Ube2D3-mediated and Ube2N-mediated RIG-I activation.

**Ubiquitination of RIG-I by Ube2D3-Riplet at K48/96/172**. On on viral infection, RIG-I was ubiquitinated, which is dependent on Riplet (Supplementary Fig. 6d). To determine the modification sites in RIG-I, the ubiquitination reaction was performed with full-length RIG-I as described (Fig. 7a). After incubation, RIG-I was isolated by IP and subjected to silver staining and mass spectrometric analysis. Multiple lysine residues of RIG-I was identified to be covalently conjugated by ubiquitin moiety (Fig. 7b). To reveal the functional relevance of ubiquitination on these sites, we mutated these residues and expressed the resultant mutant forms in HEK293T ($Rig\text{-}i^{-/-}$) cells. As expected, wild-type RIG-I could rescue the defect of IFN production in $Rig\text{-}i^{-/-}$ cells on virus infection (Fig. 7c). Strikingly, double mutations (K96/172R) showed very weak rescuing activity and triple mutations (K48/96/172R) lost rescuing activity completely. All other mutants demonstrated rescuing activity comparable to wild type RIG-I. Consistently, RIG-I-(K48/96/172R) could not be ubiquitinated in response to virus infection (Fig. 7d). As RIG-I and MAVS activation could be mediated by either Ube2D3 or Ube2N through two different mechanisms, we tested the rescuing activities of RIG-I mutants in double knockout cells HEK293T ($Rig\text{-}i^{-/-}$ and $Ube2D3^{-/-}$) or HEK293T ($Rig\text{-}i^{-/-}$ and $Ube2N^{-/-}$). Notably, none of the mutant forms could rescue IFN production in HEK293T ($Rig\text{-}i^{-/-}$ and $Ube2N^{-/-}$) cells (Fig. 7e), suggesting that all three residues K48/96/172 are critical for the ubiquitination and activation of RIG-I by the Ube2D3-Riplet pair. Interestingly, mutants (K48/96R) and (K48/172R)

**Figure 3 | Ub2D3 and Ube2N are required for RIG-I and MAVS activation on virus infection in HEK293T cells.** (**a**,**b**) The cell-free assay containing vRNA and purified components (E1, Ub, RIG-I and Riplet) was performed as described in Fig. 2b, except that S100 fraction was substituted with purified E2 recombinant proteins as indicated. Each E2 protein was analysed at three different concentrations. Lane 13 did not contain E2 proteins ( − E2). (**c**) HEK293T cells (wild-type and knockout lines) were transfected with or without si-Ube2D3 RNAs. Seventy two hours after transfection, the cells were infected with VSV. Twelve hours post virus infection, the cells were collected for measuring IFN induction by qPCR. *$P < 0.05$ and ***$P < 0.001$. (**d**,**e**) Cells were treated as described in **c**. P5 fractions were isolated to examine MAVS aggregation (**d**), and whole cell lysate were used to analyse knockdown or knockout effect in genes as indicated (**e**). The original full blot for (**e**) can be found in Supplementary Fig. 8d. (**f**) si-Ube2D3 RNAs were transfected into HEK293T ($Ube2N^{-/-}$) cells. Seventy two hours after siRNA transfection, the cells were further transfected with pcDNA-flag-Ube2D3 or pcDNA-flag-Ube2N as indicated. Twenty four hours after transfection of pcDNA plasmids, the cells were infected with VSV for another twelve hours. The cells were then collected for analysis of MAVS aggregation. See also Supplementary Fig. 3f.

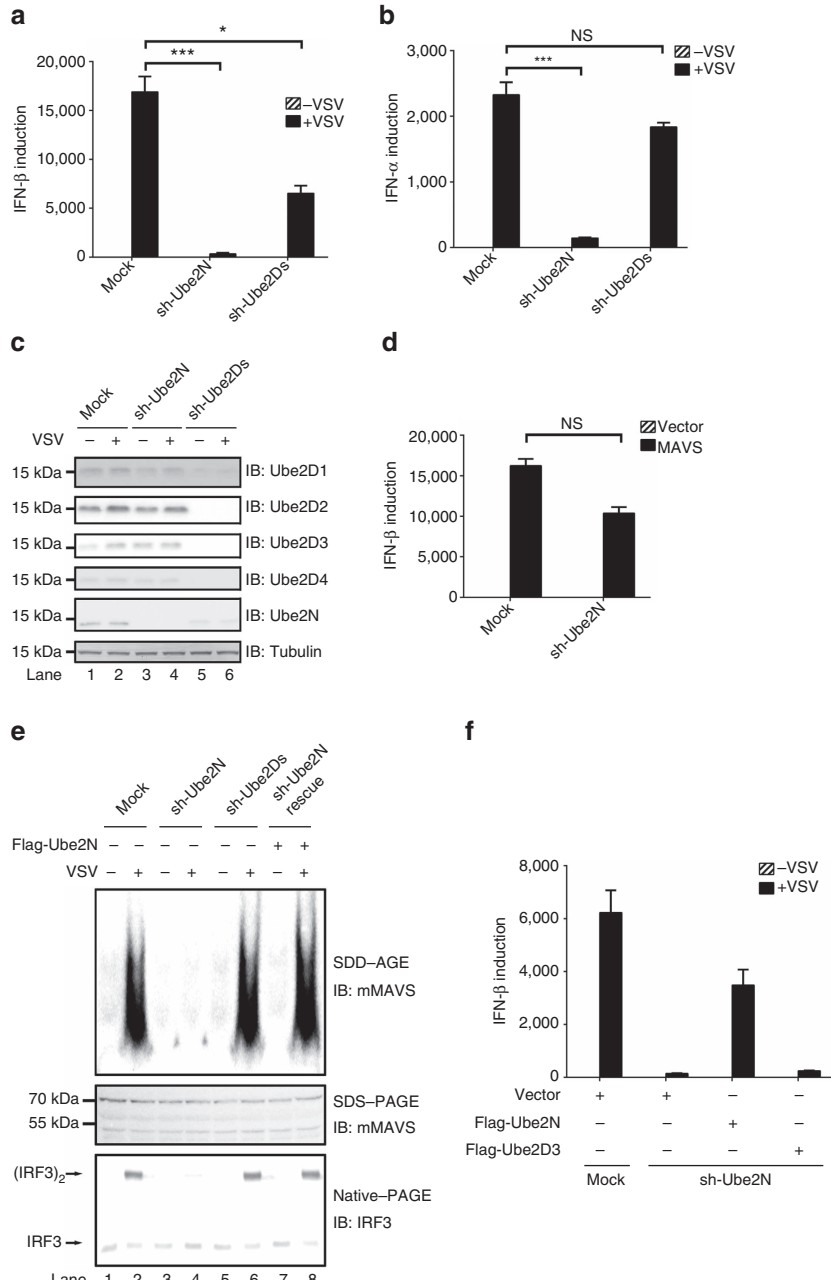

**Figure 4 | Ube2N is required for RIG-I and MAVS activation in MEFs.** (**a**–**c**) Small hairpin RNAs (sh-Ube2D1/2/3/4 & N) as indicated was transduced into MEFs with lentivirus. Eighty four hours after transduction, the cells were infected with VSV. Sixteen hours post infection, the cells were collected and IFN β (**a**) and IFN α (**b**) production were measured by qPCR. Immunoblotting were also performed to examine knockdown efficiency (**c**). *$P < 0.05$ and ***$P < 0.001$. NS indicates no statistically significant difference. (**d**) MEFs were transduced with or without sh-Ube2N. Eighty four hours after transduction, Flag-MAVS was further transduced into MEFs by retrovirus for forty eight hours. The cells were then infected with VSV. Sixteen hours post infection, the cells were collected to measure cytokine production by qPCR. See also Supplementary Fig. 4d. (**e**) MEFs were treated as described in **a** except that the cells were collected to isolate P5 fractions. P5 fractions were subjected to SDD-AGE to examine MAVS aggregation and IRF3 dimerization assay *in vitro*. The original full blot can be found in Supplementary Fig. 9a. (**f**) MEFs were treated as described in **d** except that Flag-Ube2D3 or Flag-Ube2N but not Flag-MAVS was transduced into the cells by retrovirus as indicated. See also Supplementary Fig. 4e.

could rescue IFN production in HEK293T ($Rig\text{-}i^{-/-}$ and $Ube2D3^{-/-}$) cells, while mutations (K96/172R) and (K48/96/172R) abolished the rescuing ability of RIG-I, indicating that residues K96/172 but not K48 might play an important role in RIG-I activation by the Ube2N-Riplet pair. Consistently, K96/172R mutations abolished the binding of RIG-I to polyubiquitin chains (Supplementary Fig. 7a).

## Discussion

In this report, we reconstitute a cell-free assay to study RIG-I activation, which mimics the viral RNA-triggered antiviral innate immune signalling. First, we uncover the pivotal role of E3 ubiquitin ligase Riplet in RIG-I activation. Second, by biochemical fractionation, we identify ubiquitin-conjugating enzymes Ube2D3 and Ube2N as key activators for RIG-I-mediated antiviral

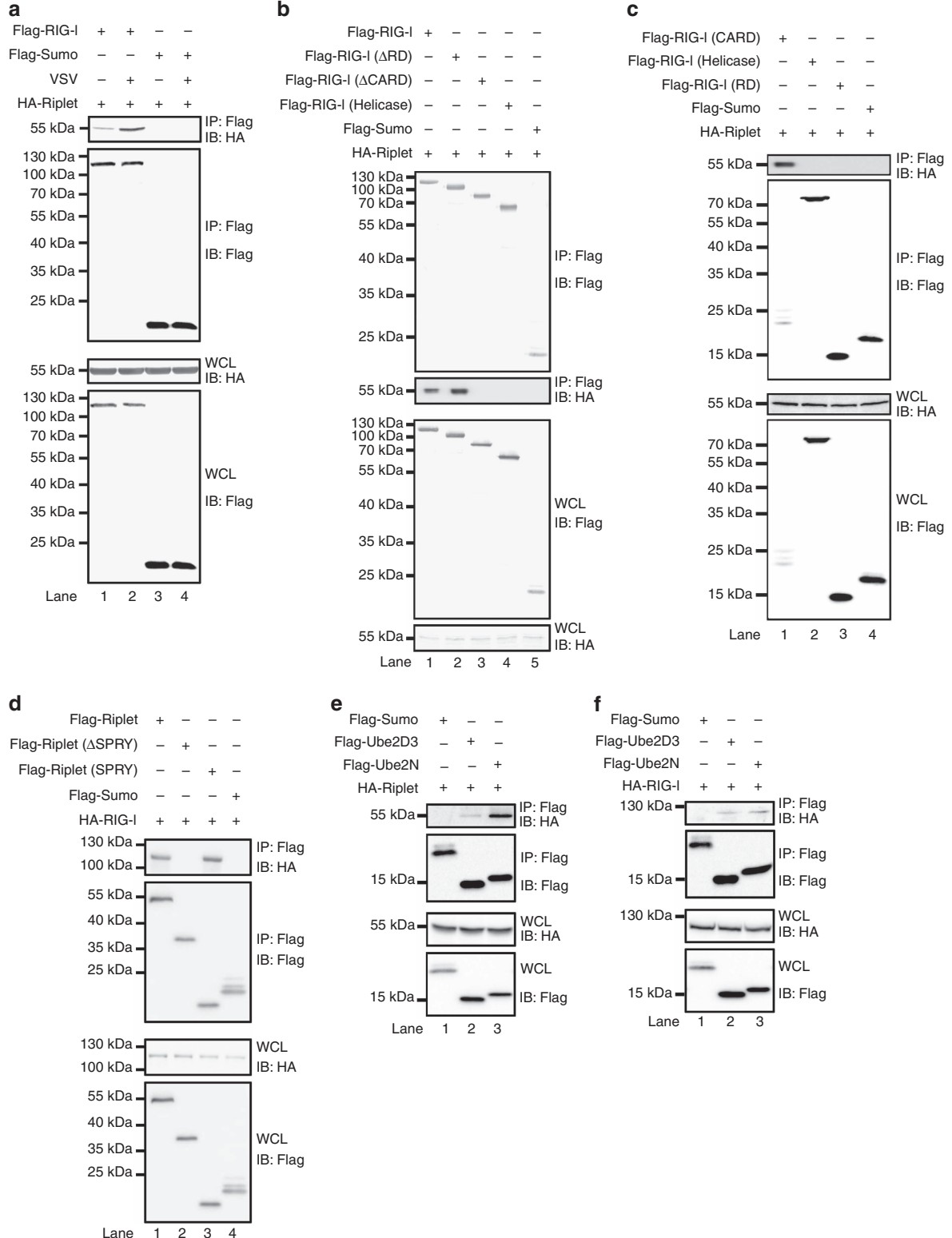

**Figure 5 | The association of RIG-I with Riplet and Ube2D3/Ube2N.** (**a**) pcDNA3-flag-sumo, pcDNA3-flag-RIG-I and pcDNA3-HA-Riplet were transfected into HEK293T cells as indicated. Twenty four hours after transfection, the cells were infected with or without VSV. The cells were collected 12 h post infection and lysed for IP with M2 beads. IP products and whole-cell lysate (WCL) were subjected to immunoblotting. The original full blot can be found in Supplementary Fig. 9b. (**b**,**c**) pcDNA3-flag-sumo, pcDNA3-flag-RIG-I (full length and deletions) and pcDNA3-HA-Riplet were transfected into HEK293T cells. Thirty six hours after transfection, IP was performed as described in **a**. RIG-I-ΔRD covers aa-1-794; RIG-I-ΔCARD covers aa-229-925; RIG-I-Helicase covers aa-229-794; RIG-I-2CARD covers aa-1-200; RIG-I-RD covers aa-795-925. (**d–f**) pcDNA3-HA-RIG-I, pcDNA3-flag-Riplet, pcDNA3-flag-sumo pcDNA3-HA-Riplet, pcDNA3-flag-Ube2D3 and pcDNA3-flag-Ube2N were transfected into HEK293T cells. Thirty-six hours after transfection, IP was performed. Riplet-SPRY covers aa 261–428.

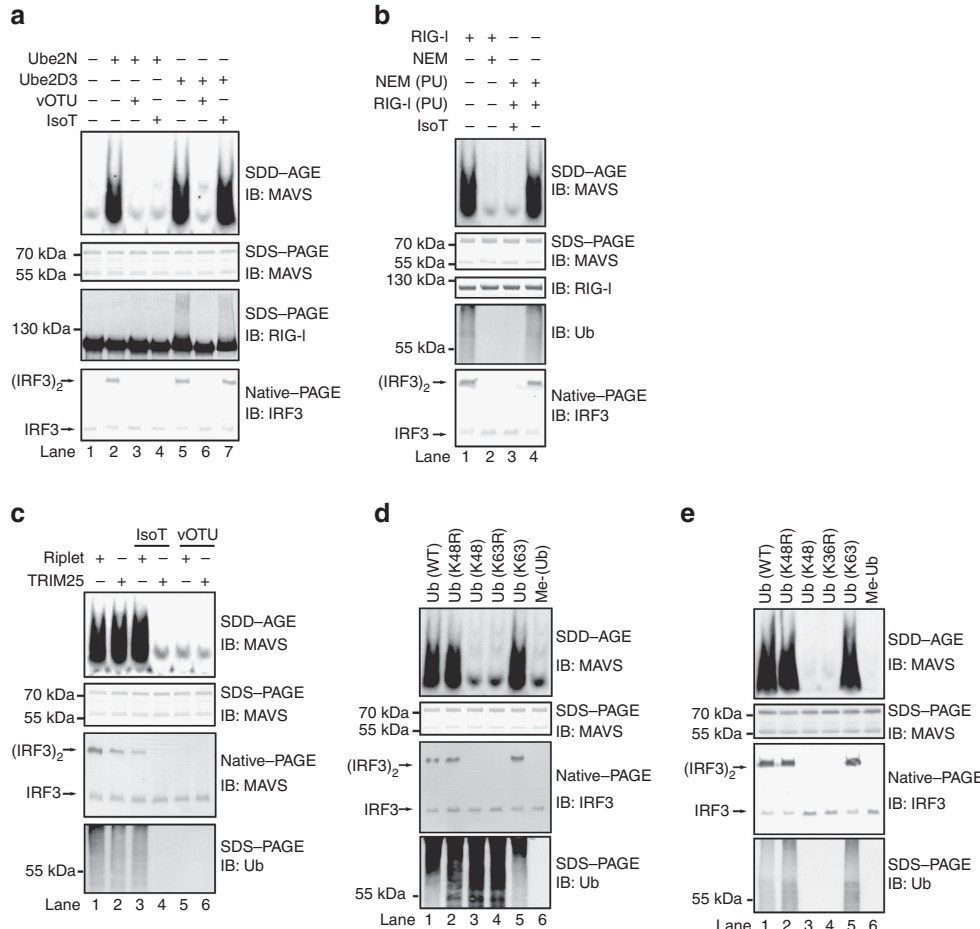

**Figure 6 | Covalently anchored and unanchored polyubiquitin chains are two alternative mechanisms for RIG-I and MAVS activation. (a)** The cell-free assay was performed using purified components (E1, Ube2N or Ube2D3, Riplet, ubiquitin, RIG-I) as described in Fig. 3b. The reaction mixtures were treated with or without IsoT or vOTU for one hour before further incubation with P5 fraction to measure MAVS aggregation and activity in stimulating IRF3 dimerization. An aliquot of the reaction mixture was subjected to immunoblotting to examine RIG-I ubiquitination. The original full blot can be found in Supplementary Fig. 9c. **(b)** The cell-free assay (containing E1, Ube2N, Riplet and ubiquitin) was performed as described in **a** in the presence (Lanes 1 and 2) or absence of RIG-I (Lanes 3 and 4). Lane 2 also included NEM. After incubation, NEM and RIG-I were added (Lanes 3 and 4). The sample in Lane 3 was further treated with IsoT. The reaction mixtures were then incubated with P5 fraction to examine MAVS aggregation. PU: post ubiquitination, indicating that NEM and RIG-I were added after the incubation of ubiquitin enzymes and ubiquitin (Lanes 3 and 4). **(c)** The cell-free assay was performed using purified components (E1, Ube2D3, Riplet or TRIM25, ubiquitin, RIG-I) as described in **a**. The reaction mixtures were treated with or without IsoT or vOTU before further incubation with P5 fraction to measure MAVS aggregation and activity in stimulating IRF3 dimerization. An aliquot of the reaction mixture was subjected to immunoblotting to examine polyubiquitin chain formation. **(d)** The cell-free assay was performed using purified components (E1, Ube2D3, Riplet, various forms of ubiquitin, RIG-I) as described in **a**. K48R: lysine48 of ubiquitin was mutated to R; K48: all lysine residues of ubiquitin except lysine48 were mutated to R; K63R: lysine63 of ubiquitin was mutated to R; K63: all lysine residues of ubiquitin except lysine63 were mutated to R; Me-Ub: methylated ubiquitin, as a negative control. **(e)** The cell-free assay was performed as described in **(d)** except that Ube2D3 was omitted and Ube2N was used.

signalling. Third, using purified components including E1, Ube2D3 or Ube2N, Riplet and RIG-I, we successfully reconstitute the activation of RIG-I and MAVS triggered by vRNA. Finally, we identify three critical residues of RIG-I for polyubiquitin chain conjugation, and thus reveal that covalent conjugation of polyubiquitin chains and unanchored polyubiquitin chains are two parallel mechanisms mediated by Ube2D3 and Ube2N respectively for RIG-I activation.

Ubiquitination plays an important role in antiviral innate immune signalling[30–36]. K63-linked polyubiquitin chains are required for RIG-I pathway in at least two processes: one is for RIG-I to activate MAVS and the other is for MAVS to activate TBK1. E3 ubiquitin ligases required for MAVS to activate TBK1 are TRAF proteins (that is, TRAF2, 3, 5, 6), which may play redundant roles[37,38]. On the other hand, TRIM25 was the first

identified E3 ubiquitin ligase that was critical for RIG-I activation and following studies uncovered that other E3s such as Riplet, MEX3C and TRIM4 are indispensable for RIG-I activation. We here demonstrated Riplet, but not TRIM25 or other E3 ligases, is specifically required for Rig-I activation. Nevertheless, we cannot rule out the possibility of the involvement of other E3s abovementioned in RIG-I pathway under certain circumstances such as some specialized cell type.

RIG-I pathway is composed of multiple steps, so that a result is prone to being complicated by the difficulty in pinpointing the exact step involved and ruling out the possibility of indirect effect. Our cell-free assay using defined components avoids this issue, and thereby allows the precise dissection of key mechanisms. With this cell-free assay, we confirmed that Riplet is indispensable for RIG-I activation. Furthermore, we identified two

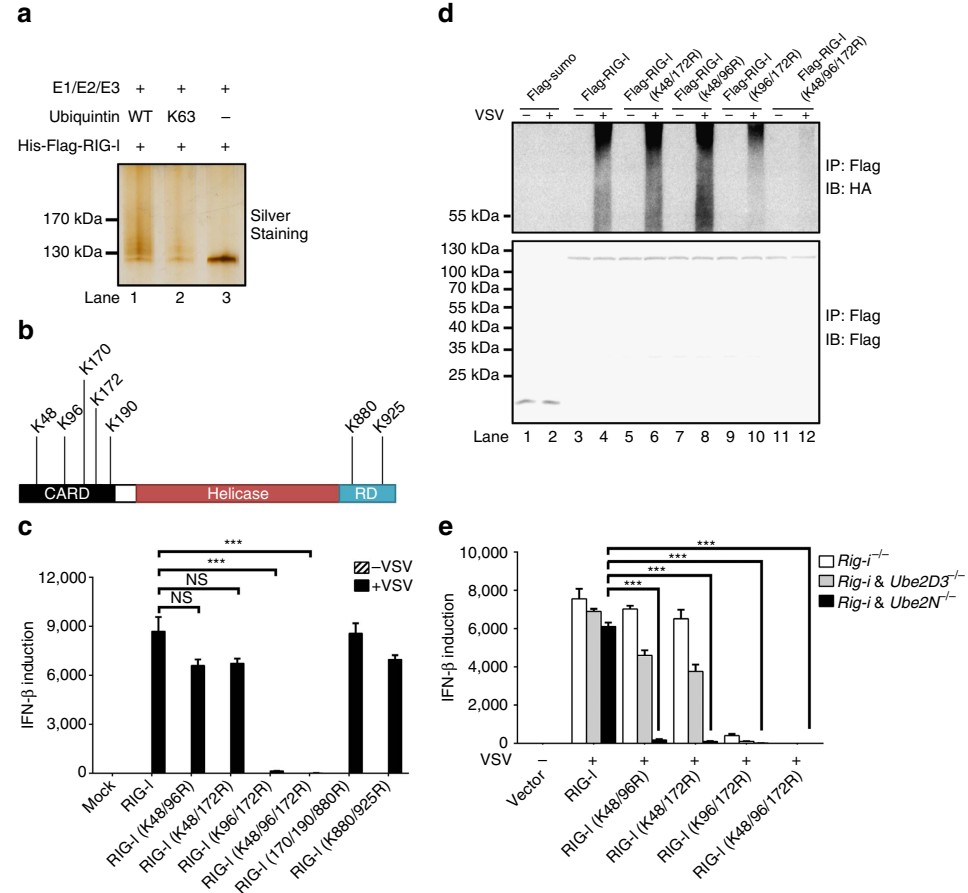

**Figure 7 | Modification sites of RIG-I by polyubiquitin chains.** (**a**) The ubiquitination reaction was performed using purified components (E1, Ube2D3, Riplet, ubiquitin (WT or K63), His-flag-RIG-I). His-flag-RIG-I was then immunoprecipitated by M2 beads and eluted with flag peptide, followed by SDS-PAGE and silver-staining. (**b**) Ubiquitination sites of RIG-I identified by mass spectrometric analysis. (**c**) pcDNA3-flag-RIG-I (wild type and various mutant forms) were transfected into HEK293T ($Rig\text{-}i^{-/-}$) cells. Thirty six hours after transfection, the cells were infected with or without VSV for twelve hours, which then were collected to measure IFN induction by qPCR. *$P < 0.05$ and ***$P < 0.001$. (**d**) pcDNA3-HA-ubiquitin and pcDNA3-flag-sumo or pcDNA3-flag-RIG-I (wild type and various mutant forms) were transfected into HEK293T ($Rig\text{-}i^{-/-}$) cells. Thirty six hours after transfection, the cells were infected with or without VSV. Twelve hours post infection, the cells were collected for IP with M2 beads and IP products were subjected to immunoblotting. The original full blot can be found in Supplementary Fig. 9d. (**e**) Wild type and various mutant forms of RIG-I were transfected into HEK293T ($Rig\text{-}i^{-/-}$), HEK293T ($Rig\text{-}i^{-/-}$ & $Ube2D3^{-/-}$) or HEK293T ($Rig\text{-}i^{-/-}$ & $Ube2N^{-/-}$) cells. Thirty six hours after transfection, the cells were infected with VSV for twelve hours, which then were collected to measure IFN induction by qPCR. *$P < 0.05$ and ***$P < 0.001$.

essential regulators for RIG-I activation, that is, Ube2D3 and Ube2N, through biochemical purification. Notably, RIG-I activation can be reconstituted with purified components, including RIG-I, Riplet, Ube2D3 or Ube2N, E1 and ubiquitin, in the cell-free assay. Our findings are consistent with previous reports showing that Ube2D3 and Ube2N are critical for RIG-I activation in siRNA-mediated knock-down experiments[25,37]. We provide several lines of evidence to show Ube2D3 and Ube2N play redundant roles in RIG-I activation in human cells. Interestingly, we found that Ube2N but not Ube2D3 plays a critical role in RIG-I and MAVS activation in multiple mouse primary cells, indicating the divergence in antiviral innate immune signalling across species.

Currently, how ubiquitination mediates RIG-I activation remains controversial. Two mechanisms have been proposed to regulate RIG-I activation, that is, covalent conjugation of polyubiquitin chains and unanchored polyubiquitin chains. Unanchored polyubiquitin chains requires non-covalent binding to RIG-I[25]. Nevertheless, it is critical to employ physiologically relevant E2(s) and E3(s) to investigate the mechanisms underlying RIG-I activation by polyubiquitin chains. With genuine E3 and E2s determined in this study, we investigate the

unresolved mechanisms. We demonstrate that Ube2D3-Riplet pair can conjugate polyubiquitin chains to RIG-I, while Ube2N-Riplet pair preferentially makes unanchored polyubiquitin chains. Both conjugated (to RIG-I) and unanchored polyubiquitin chains are capable of activating RIG-I, which is conceivable in light of that conjugated polyubiquitin chains could activate RIG-I *in trans*. As a matter of fact, this view is consistent with the structural study showing that conjugated polyubiquitin chains to RIG-I 2CARD is compatible with their non-covalent binding to 2CARD, which may promote the formation of 2CARD tetramer to assemble MAVS CARD filament[27]. We also identify three critical residues (K48/96/172) of RIG-I for polyubiquitin chain conjugation and its activation by Ube2D3-Riplet pair. In contrast, residues K96/172 but not K48 are important for RIG-I activation by Ube2N-Riplet pair. Collectively, we conclude that covalent conjugation and non-covalent binding of polyubiquitin chains to RIG-I are two alternative mechanisms for its activation, dictated by two ubiquitin-conjugating enzymes Ube2D3 and Ube2N, respectively. Whether there is a preferential selection of either E2s for innate immune signalling in the context of various cell types or virus species awaits future investigation.

## Methods

**Plasmids and antibodies.** Human complementary DNA was amplified from HEK293T cells. cDNAs of *Riplet*, *Trim25*, *Ube2D1*, *Ube2D2*, *Ube2D3*, *Ube2D4*, *Ube2E1*, *Ube2M*, *Ube2L3*, *Ube2N* and *Ube2V1* were cloned into pcDNA3-flag expression vector and primers used for the amplification can be found in Supplementary Table 2. For retrovirus-mediated transduction, N-terminally Flag-tagged MAVS, TRIM25, RIG-I, Ube2N and Ube2D3 were cloned into the expression vector pMX between restriction enzyme sites of *Bam*HI and *Not*I. All mutations and deletions were made with the Fast-mutagenesis Kit (TransGen Biotech, Beijing, China), QuickChange Lightning Multi Site-Directed Mutagenesis Kit (Stratagene) or overlapping PCR strategy. All constructs were confirmed by DNA sequencing. Antibodies against human MAVS, mouse MAVS and human Riplet were raised by immunizing rabbits with recombinant proteins His-sumo-hMAVS-(aa-301-460), His-sumo-mMAVS-(aa-101-434) and His-sumo-Riplet (full length). Commercial antibodies included anti-Flag (Sigma, F3165, F7425, dilution 1:5,000), anti-tubulin (Sigma, T5168, dilution 1:7,500), anti-HA (Cell Signaling Technology, 3724S, dilution 1:2,000), anti-RIG-I (Cell Signaling Technology, 3743, dilution 1:1,000), anti-Ube2N (Cell Signaling Technology, 4919, dilution 1:1,000), anti-ubiquitin (Santa Cruz Biotechnology, sc-8017, dilution 1:1,000), anti-Ube2E1 (Santa Cruz Biotechnology, sc136113, dilution 1:1,000), anti-TRIM25 (Abcam, ab167154, dilution 1:10,000), anti-IRF3 (Abcam, 2241-1, dilution 1:3,000), anti-UBA1 (Abcam, ab181225, dilution 1:5,000), anti-Ube2D1 (Abcam, ab176561, dilution 1:10,000), anti-Ube2D2 (Abcam ab155088, dilution 1:10,000), anti-Ube2D3 (Abcam, ab176568, dilution 1:4,000) and anti-Ube2D4 antibodies (Abcam, ab179401, dilution 1:1,000).

**Cells and viruses.** MEF cells, PEM cells (peritoneal macrophages, isolated from wild-type C57BL/6 mouse) and BMDM cells (Bone Marrow-Derived Macrophages, isolated from wild-type C57BL/6 mouse and induced by M-CSF for seven days) were grown in DMEM supplemented with 10% fetal bovine serum (FBS, ExCell Bio, FSP500), penicillin (100 U ml$^{-1}$) and streptomycin (100 µg ml$^{-1}$). HEK293T cells were cultured in DMEM medium supplemented with 10% calf serum and antibiotics. Recombinant virus VSV-ΔM51-GFP was amplified in vero cells and used with a multiplicity of infection (MOI) of 1. Sendai virus was a kind gift from Dr Xiaodong Zhang (Wuhan University) and used at a concentration of 50 HA units per ml.

**Subcellular fractionation and IRF3 dimerization assay.** Subcellular fractionation was performed for HEK293T or MEFs as described[14]. In brief, cellular lysate in hypotonic buffer ( Buffer A: Tris-Cl 10 mM pH 7.5, KCl 10 mM, EGTA 0.5 mM and MgCl$_2$ 1.5 mM) was obtained and centrifuged at 1,000 g for 5 min. The supernatant (S1) was isolated and further centrifuged at 10,000 g for 10 min. The supernatant (S5) and pellet (P5) were then separated. S5 was subjected to centrifugation at 100,000 g for 30 min, and the resultant supernatant (S100) was collected. P5 was treated with RIG-I and Ub mixture as indicated, which was then isolated and incubated with S5 in the presence of ATP, followed by native gel electrophoresis to examine IRF3 dimerization as described[14].

**Ubiquitination-mediated RIG-I and MAVS activation in the cell-free assay.** Initial cell-free assay (related to Figs 1a and 2a) contained S1 fraction (protein concentration 15 µg µl$^{-1}$), His-RIG-I (full-length or 2CARD) or His-sumo -Riplet (wild type or mutant form) 20 ng µl$^{-1}$, vRNA 50 ng µl$^{-1}$ and ATP 2 mM, which were incubated for 1 h at 30 °C. The reaction mixture was further incubated with 1 µl P5 at room temperature for 1 h, followed by centrifugation at 10,000 g for 5 min. The pellet (i.e., activated P5) was subjected to immunoblotting or used for IRF3 dimerization assay. Similarly, refined cell-free assay (related to Fig. 2b) was performed except that S1 was replaced by S100 fraction and E1 20 ng µl$^{-1}$, ubiquitin 50 ng µl$^{-1}$, RIG-I 20 ng µl$^{-1}$ and Riplet 20 ng µl$^{-1}$ were supplemented. In the cell-free assay using purified components (related to Fig. 3a,b), reaction mixture (20 µl) contained 10 ng His-E1, 15 ng His-E2 (Ube2D1, Ube2D2, Ube2D3, Ube2D4 or Ube2N), 50 ng E3 (His-sumo-Riplet or His-TRIM25), 1 µg ubiquitin, 1 µg vRNA and 50 ng His- RIG-I in Buffer B (Tris-Cl 20 mM pH 7.5, ATP 2 mM, MgCl$_2$ 5 mM and DTT 0.5 mM). After incubation for 1 h at 30 °C, the reaction mixture was further incubated with P5 as described above.

**Purification of endogenous E2s required for the cell-free assay.** HEK293T *Riplet*$^{-/-}$ cells were collected and resuspended in Buffer A. The cells were broken by electronic homogenizer and centrifuged at 100,000 g for 30 min to get S100. S100 fraction was loaded onto Mono-Q sepharose column (GE healthcare) pre-equilibrated with Buffer C (Hepes 20 mM pH 7.5, DTT 1 mM and PMSF 1 mM) and the flow-through fraction was collected as Fraction A. The column was then washed with Buffer D (Hepes 20 mM pH 7.5, NaCl 0.5M, DTT 1 mM and PMSF 1 mM) and Fraction B was collected. Fraction C was eluted with Buffer E (Hepes pH 7.5 20 mM, NaCl 1M, DTT 1 mM and PMSF1 mM). Fraction A was loaded onto Heparin column (GE healthcare) preequilibrated with Buffer C, and active fractions were eluted with buffer Buffer E, which were pooled and subjected to buffer-exchange to Buffer F (NaAc 20 mM pH 6.0, DTT 1 mM and PMSF 1 mM). Active fractions in Buffer F were loaded onto mini-S sepharose column (GE healthcare). Active fractions from mini-S column were eluted with Buffer G

(NaAc 20 mM pH 6.0, NaCl 1 M, DTT 1 mM and PMSF 1 mM), which were loaded onto a Superdex200 column (GE healthcare) preequilibrated with Buffer H (Hepes 20 mM pH 7.5, NaCl 150 mM, DTT 1 mM and PMSF 1 mM). Fractions from Superdex200 column were subjected to SDS–PAGE and the cell-free assay to examine RIG-I and MAVS activation. All procedures were carried out at 4 °C by using ÄKTA or ÄKTAmicro system.

**NEM and PYR-41 treatment.** NEM and PYR-41 were dissolved in ethanol and DMSO, respectively. In cell-free assay (related to Fig. 1b), both NEM and PYR-41 were added at a final concentration of 40 µM. To treat cells (related to Fig. 1c), PYR-41 was added to the medium at a final concentration of 15–45 µM. In Fig. 6b, the cell-free assay was carried out in a reaction mixture of 100 µl containing 1 µg E1, 3 µg E2 (Ube2D3 or Ube2N/Ube2V1), 3 µg Riplet, and 30 µg ubiquitin in Buffer I (Hepes pH 7.5 20 mM, 2 mM ATP, 5 mM MgCl$_2$ and 0.5 mM DTT). After incubation for 3 h at 30 °C, 10 mM DTT was added to the mixture, which was kept at room temperature for 20 min. NEM was then added to a concentration of 40 mM. After another incubation for 20 min, DTT was added to a final concentration of 15 mM to quench excessive NEM. The reaction mixture was switched to Buffer J (Hepes pH 7.5 20 mM, 10% Glycerol, 50 mM NaCl, 1 mM DTT and 1 mM PMSF) by repeated dilution and centrifugation.

**Generation of knock-out cell lines in HEK293T and MEFs.** *Rig-i*$^{-/-}$, *Riplet*$^{-/-}$, *Trim25*$^{-/-}$, *Mex3c*$^{-/-}$, *Trim4*$^{-/-}$, *Ube2N*$^{-/-}$, *Rig-i* and *Riplet* double deficient, *Ube2D1*$^{-/-}$, *Ube2D2*$^{-/-}$, and *Ube2D4*$^{-/-}$ and Ube2D1/2/4 and Ube2E1 and Ube2N$^{-/-}$ HEK 293T cells were generated by CRISPR/Cas9 technique and the sg-RNA sequence can be found in Supplementary Table 2. *Rig-i*$^{-/-}$, *Riplet*$^{-/-}$ and *Trim25*$^{-/-}$ MEFs were also generated by CRISPR/Cas9 technique. Briefly, sg-RNA for each gene were cloned into a CRISPR/Cas9-based vector pX330 with a puromycin resistance selection marker. These vectors were transfected into HEK293T cells or MEFs by Lipofectamine 2000 (Invitrogen). After selection with puromycin (0.5 µg ml$^{-1}$), single-colonies were picked and verified by genome sequencing and immunoblotting.

**Real-time PCR.** The procedure was as described[14]. Briefly, total RNA was extracted using the RNA simple total RNA kit (Tiangen, Shanghai, China) from HEK 293T or MEF cells, and the reverse transcription was performed using the GoScript Reverse Transcription system (Promega). cDNAs were then used as templates for qPCR assay using SuperReal Premix Plus (Tiangen). As internal controls, GAPDH and β-actin were used for HEK 293T and MEF cells respectively. Induction fold was determined with the ΔΔCq method and qPCR Primers used to amplify specific genes were showed in Supplementary Table 2.

**Treatment with siRNA and shRNA.** siRNA sequence targeting *Ube2D3* can be found in Supplementary Table 2. si-Ube2D3 was transfected into HEK293T cells as described[15]. To knockdown gene expression in MEF cells, we used PLKO.1-based lenti-viral transduction and the sequence of shRNA can be found in Supplementary Table 2. Lenti-viral vector and viral packaging vectors were transfected into HEK293T cells using Lipofectamine 2000 (Invitrogen). Forty eight hours later, culture medium containing packaged lenti-viral particles were collected and used to infect MEF cells. MEF cells were incubated with lentivirus in the presence of 8 µg ml$^{-1}$ polybrene for 24 h before fresh medium was served. Thirty six hours after infection, puromycin (2 µg ml$^{-1}$) was added to the culture medium to select cells stably expressing the shRNA transduced. After another 48 h, the cells were used for following experiments. For gene knockdown in PEM and BMM cells, siRNA was transfected to cells with lipofectamine 2000 (Invitrogen). Forty eight hours later, cells were infected with recombinant virus VSV-ΔM51-GFP for 6 h, and then collected for total RNA extraction and real-time PCR analysis.

**Recombinant protein preparation.** The pFastBacHT B vector and Tn5 cells were provided by Dr Degui Chen (Shanghai Institute of Biochemistry and Cell Biology). His-flag-RIG-I and His-TRIM25 were cloned into pFastBacHT B vector. The constructs were transformed into DH10Bac competent cells to get bacmids. The bacmids were then transfected into Tn5 cells using cellfectin (Invitrogen). The supernatants containing baculovirus were collected and used for the following infection to amplify virus. After infection for 68 h, cells were collected and lysed by sonication. Recombinant proteins were purified by Ni-NTA beads. Wild type or mutant forms of Ube2D1, Ube2D2, Ube2D3, Ube2D4, Ube2E1, Ube2N, Ube2M, Ube2L3 and ubiquitin were cloned into pET14b with an N-terminal 6XHis tag and expressed in BL21 strain (PJY2), which were induced by 0.2 mM isopropyl-b-D-thiogalactoside (IPTG) for 4 h at 37 °C. pET28a-His-sumo-Riplet WT and its dominant negative form were transformed into BL21 (TransGen Biotech) and their expression were induced by 0.2 mM IPTG for 9 h at 18 °C. IsoT and vOTU were cloned into pGEX (GE healthcare) with a N-terminal GST tag and expressed in BL21 at 18 °C for 6 h. Bacterial cultures were collected and sonicated in Buffer K (10 mM Tris-Cl pH 7.5, 0.3 M NaCl, 0.5 mM DTT and 1 mM PMSF). The recombinant proteins were purified by Ni-NTA beads or Glutathione-Sepharose affinity gel and further polished by size-exclusion chromatograph.

**Mass spectrometric analysis.** Protein gel bands were de-stained, reduced with 10 mM DTT and then alkylated with 55 mM iodoacetamide, followed by digestion with sequencing grade trypsin (10 ng µl$^{-1}$ in 50 mM ammonium bicarbonate) overnight at 37 °C. Peptides were extracted with 5% formic acid/50% acetonitrile and 0.1% formic acid/75% acetonitrile sequentially and then concentrated to ∼20 µl. The extracted peptides were separated by a homemade C18 analytical capillary column (50 µm × 10 cm, 5 µm resin) on a Waters nanoAcquity UPLC system (Waters, Milford, USA). The eluted peptides were sprayed into a LTQ Orbitrap Velos mass spectrometer (ThermoFisher Scientific, USA) operated in data-dependent mode with one MS scan followed by ten HCD (High-energy Collisional Dissociation) MS/MS scans for each cycle. Database searches were performed on an in-house Mascot server (Matrix Science Ltd., London, UK). The search parameters are: 7 p.p.m. mass tolerance for precursor ions; 0.02 Da mass tolerance for product ions; the following variable modifications were included: oxidation on methionine, carbamidomethylation on cysteine, and ubiquitination on lysine.

**SDD-AGE and retroviruses packaging.** Crude mitochondria (P5) were isolated from HEK293T cells or MEFs and resuspended in 1 × sample buffer (0.5 × TBE, 10% glycerol, 2% SDS, and 0.0025% bromophenol blue) and loaded onto a vertical 1.5% agarose gel. After electrophoresis in the running buffer (1 × TBE and 0.1% SDS) for 30 min with a constant voltage of 100 V at 4 °C, western immunoblotting was performed[14].

**Immunoprecipitations.** HEK293T cells were transfected with indicated expression vectors. 36 h later, cells were collected and lysed in Buffer L (HEPES 20 mM pH 7.5, NaCl 150 mM, Triton X-100 1%, DTT 1 mM and PMSF 1 mM). Cellular lysate was incubated with anti-flag M2 affinity gel (Sigma A2220) at 4 °C for 4 h. M2 beads were then spun down and washed three times with Buffer L. IP products were eluted by 50 ng µl$^{-1}$ 3XFlag peptides at 4 °C for 30 min for the following assay, or directly subjected to SDS–PAGE and immunoblotting.

**Statistical analysis.** All data are presented as the mean values based on three independent experiments, and error bars indicate s.d. Statistics significance between two groups was determined by two-tailed Student's $t$-test. A $P$ value of $< 0.05$ was considered statistically significant.

**Data availability.** The data that support the findings of this study are available from the corresponding author on reasonable request.

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

## Acknowledgements

We are grateful to Dr Zhijian 'James' Chen (University of Texas Southwestern Medical Center at Dallas, USA) for initial reagents. We thank Dr Xuewu Zhang (University of Texas Southwestern Medical Center at Dallas, USA) and Dr Shao-Cong Sun (University of Texas MD Anderson Cancer Center) for critical comments on the manuscript. We thank Dr Xiaodong Zhang (Wuhan University) for providing Sendai virus, Dr Zongping Xia (Zhejiang University) for providing constructs expressing vOTU and IsoT, and Dr Degui Chen (SIBCB) for providing pFastBacHT B vector and Tn5 cells. We thank Dr Hongyan Wang and Ms Ying Qiu (SIBCB) for help with mouse primary cells. We thank Dan Zhang and Xiaoyan Li for technical assistance. This work was supported by grants from the National Natural Science Foundation of China (31470867), the Strategic Priority Research Program of the Chinese Academy of Sciences (XDB19000000) and the CAS/SAFEA International Partnership Program for Creative Research Teams.

## Author contributions

Y.S., B.Y., W.Z. and R.Z. conducted experiments. L.L. and S.C. did mass spectrometric analysis. X.H. helped with molecular cloning. F.H. organized the study and prepared the manuscript. All authors discussed the results and commented on the manuscript.

## Additional information

**Competing interests:** The authors declare no competing financial interests.

