## [Peer Review File · Nature Communications]

Reviewers' Comments:

Reviewer #1 (Remarks to the Author):

This manuscript reports two important findings that (1) Riplet, not Trim25, is the E3 ligase that activates RIG-I, (2) Riplet functions together with the two distinct E2 enzymes, Ube2D3 and Ube2N, to use both unanchored and anchored Ub chains for RIG-I activation. The authors used cell free assays to dissect the role of individual E3 ligases in RIG-I activation. Subsequent biochemical fractionation analysis led to the identification of Ube2D3 and Ube2N as the partner E2 enzymes that work with Riplet. These findings are novel and important, and are well supported by the data. However, there are a few areas that need further clarification and additional experiments to avoid confusions.

1. While the authors' findings are intriguing, they are also contradictory to some of the previous reports (by J.U. Jung, Michaela Gack and James Chen) suggesting the major role of Trim25 in RIG-I activation. The authors should address why this discrepancy might have occurred and what the authors think is the actual role of Trim25 in the RIG-I signaling pathway. Even within this manuscript, somewhat confusing/contradicting data have been presented - for example, Fig. 6c shows that Trim25 can replace Riplet in MAVS aggregation, at least in this in vitro assay.

2. Fig. 2c: the text does not match the legend. Data in Fig. 2c shows that you need both fractions A & B to stimulate MAVS aggregation. However, the text (in line 169-174 on page 9) suggests that the fraction A is sufficient. This is more confusing as it was followed by the statement that the fraction B contains E1, which is clearly necessary for RIG-I activation.

3. Fig 2c: Ube2N is thought to function as a complex with Ube2V1. Which fraction contains Ube2V1? Does the fraction requirement supports the notion that Ube2N needs Ube2V1?

4. In general, the relationships among the E2 enzymes in Fig 3 were not clear. There are apparent discrepancies among the results in Figs 3 & 4 on the role of Ube2D3, Ube2E1 and Ube2N. For example, Fig. 3b suggests that Ube2E1 is sufficient to activate MAVS, but Fig3d suggests that it is Ube2N & Ube2D3 that are important. If results in Fig 3b are due to the usage of enzymes at non-physiologically high concentration, is Fig 3b necessary? Another confusing part is the role of Ube2N. In Fig 3c, KO of Ube2N moderately reduced the IFN induction, but further deletion of 2D1,2,4 & E1 restored the IFN induction activity. In Fig. 4a, KD of Ube2N alone was sufficient to abolish IFN induction. How can the authors reconcile all these results?

5. Were the binding assays in Fig. 5b and 5c done with or without virus infection (or with viral RNAs)? If not, the experiment should be repeated with viral RNAs (the condition where native Riplet-RIG-I interaction is expected to occur).

6. The authors used cell free assay results in Fig 6D & E to argue that Riplet makes K63-linked Ub chains. This should be also shown by cellular assays (for example by comparing RIG-I ubiquitination in Riplet-deficient and -sufficient cells with linkage specific antibody against Ub).

7. The proposed roles of anchored and unanchored Ub chains seem similar to what was previously described in the report of the CARD-Ub complex structure (Peisley et al Nature 2014). The results in Fig 7 and the role of Ub should be discussed in the context of the structural mechanism of Ub mediated oligomerization of CARD.

8. Fig 7e. In the absence of Ube2D3, Ub synthesis and RIG-I activation is expected to be mediated by Ube2N (unanchored Ub chains). K96 and K172 are the conjugation sites, and are not involved CARD-Ub binding. However, Fig 7e shows that K63/172R RIG-I has no activity in Ube2D3 KO. What is the explanation of the requirement for K96 and K172 in a Ube2N-mediated activation of RIG-I?

Reviewer #2 (Remarks to the Author):

To date, several studies have shown that ubiquitin-dependent processes play a critical role in coordinating the assembly of functional signaling complexes both upstream and downstream of MAVS. Upstream MAVS, K63-linked polyubiquitin chains have been shown to regulate RIG-I activation. TRIM25, RIPL1, TRIM4 and MEX3C are all E3 ligases that have been implicated in the K63-linked ubiquitination of RIG-I (the KD and/or KO of each of the above-mentioned E3 ligases has been shown to significantly reduce RIG-I-dependent IFN induction). In addition, in vitro cell-free experiments showed that unanchored K63-linked polyUb chains generated by TRIM25 are also able to activate RIG-I signaling. However, how different E2 and E3 enzymes work together to modulate signal transduction in different species and cell types, and how different stimuli can regulate the activity of such enzymes during infection still remain to be elucidated.

In this study, the authors take advantage of in vitro assays to dissect the early events of RIG-I activation. The authors show that in their cellular systems (293Ts and MEFs), RIPL1 is the only E3 ligase that is required for MAVS aggregation as well as downstream signaling upon viral infection. By performing biochemical fractionation coupled with mass spec analysis, the authors then identify Ube2D3 and Ube2N as the two critical E2 enzymes required for Riplet-dependent RIG-I activation. Ube2N being essential for MAVS aggregation in MEFs, whereas Ube2D3 and Ube2N playing redundant roles in 293Ts. Based on data from cell-free assays, the authors then suggest that while the Riplet-Ube2N pair preferentially catalyzes the synthesis of unanchored K63-linked polyUb chains, the Riplet-Ube2D3 pair promotes covalent conjugation of polyUb chains to RIG-I.

These findings are important to better understand the molecular mechanisms regulating RIG-I activation by Riplet. However, some of the data in this manuscript are not in agreement with the pre-existing literature, and more convincing data should be provided to support them. Additional

experiments in more relevant cellular systems are needed to confirm the relevance of these findings.

Specific issues are detailed below:

- 1) In Figure 1d, Riplet CRISPR KO cells show a dramatic defect in MAVS aggregation. Did the authors check whether Riplet depletion affects the expression levels of RIG-I and other known critical proteins in the RIG-I activation cascade such as TRIM25, MAVS etc.?
- 2) In Figure 1e, TBK1 KO cells should be included as control. A WB with the expression levels of MAVS should also be included.
- 3) In Figure 1f, the authors utilize CRISPR KO MEFS that were not previously characterized. In contrast with what has been observed by several other groups, TRIM25 depletion did not affect IFN induction upon viral infection. Can the authors provide WB for these cell lines and comment on these differences? Were MEFS reconstituted with human or mouse TRIM25/RIPLET? Please specify.
- 4) A WB for the DKO cells used in figure 2B should also be provided.
- 5) In Figure 3c, Ube2N KO cells exhibited a better phenotype than Ube2D3, N&E1 KO cells. Can the authors comment on this? Also, the differences in IFN induction in Ube2N KO cells do not correlate with the differences in MAVS aggregation shown in figure 3d.
- 6) Are the differences observed in the Ube2N and Ube2D3 requirement between 293T and MEFS species-specific or cell type specific?
- 7) A role for Ube2D3 (Ubc5c) and Ube2N (Ubc13) in the RIG-I-dependent MAVS activation was previously shown by Zeng et al., 2009; Zeng et al. 2010; and Sanchez et al., 2016. This information should be included in the manuscript and discussed.
- 8) In figure 5, the authors address the interaction between Riplet, RIG-I, Ube2D3 and Ube2N by co-ip experiments in 293T cells and show that the 2CARD domain of RIG-I and the SPRY domain of RIPLET are required for their interaction. Similar co-ips studies were previously performed (Gao et al., 2009; Oshiumi et al., 2009; Oshiumi et al., 2013) and different domains appeared to be critical for RIG-I/RIPLET interaction (Oshiumi et al., 2009; Oshiumi et al., 2013). The authors should comment on the differences observed in these studies. A RIPLET w/o SPRY domain control should be included in panel 5d.
- 9) The critical role of RIPLET, Ube2D3 and Ube2N in regulating RIG-I activation should be studied in more relevant cell types such as macrophages and DCs that are the main source of Type I IFNs in vivo. The role of TRIM25 in primary cells should also be determined. Riplet ko (Oshiumi 2010) and Ube2N conditional Ko mice (Yamamoto et al., 2006) have been previously described. Primary cell lines from these mice could be used to address this issue. Figure 5 should be also be complemented with endogenous co-ip studies in more relevant cells.
- 10) Figure 7 does not convincingly prove that residues 48/96/172 are indeed covalently ubiquitinated by the riplet-ube2d3 pair in cells upon VSV infection.

Reviewer #3 (Remarks to the Author):

RIG-I-like receptors bind viral RNA and initiate the antiviral immune response through their interaction with mitochondrial antiviral signaling protein MAVS. MAVS on the mitochondrial membrane forms prion-like aggregates that robustly activate downstream kinases and

transcription factors. RIG-I activation can be regulated by the K63-linked polyubiquitination mediated by TRIM25 and Riplet E3 ligases. In addition, a critical role of unanchored K63 chains on the RIG-I activation was also shown. In this study the authors utilized a cell-free assay for viral RNA-triggered activation of RIG-I and MAVS aggregates and demonstrated that Riplet is the only E3 ligase that is required for activation of RIG-I to form MAVS aggregates. They also showed that Riplet-Ube2D3 (Ubc5c) promotes covalent conjugation of K63 chains to RIG-I while Ube2N (Ubc13) rather generates unanchored K63 polyubiquitin chains. Overall, biochemical data using a cell-free assay are convincing showing a nice correlation between MAVS aggregates and dimerization of IRF-3. However, new additional mechanistic information seems to be limited. Comments are below.

Fig. 1f. Reduction of IFN- β production was not observed in trim25 knockout cells, which is inconsistent to the previous findings (Gack et al., 2007). Verify these findings by reconstituting Flag-TRIM25 into the trim25 knockout cells. The authors may also try trim25 shRNA or siRNA.

Fig. 3. All of Ube2Ds, Ube2D1-D4, are able to activate RIG-I generating MAVS aggregates in 3a. In addition, three different concentrations of E2 are almost equally capable of activating MAVS and IRF-3. Although the authors showed that dual disruption of Ube2Ds and Ube2N abrogated the IFN induction (3c) and MAVS aggregation (3f), authors should show whether ectopic expression of Ube2D1, 2, 4 into the Ube2D1,2,4-/-&D3 siRNA cells did not restore the IFN induction and MAVS aggregates to verify the specific role of Ube2D3 in RIG-I activation. In 3c, IFN production in UbeD1,2,4&N, E1-/- cells was twice higher than those of UbeD1,2,4 -/-. Why? Is it reproducible?

Fig. 4. In a cell-free assay, Ube2Ds and Ube2N play redundant roles in MAVS aggregation. In Fig. 4, authors showed that Ube2N only play an essential role for RIG-I mediated MAVS activation. It is confusing. The authors should describe better differences in systems. In addition, please clarify whether sh-Ube2D 3 or sh-Ube2D1/2/3 is applied to Fig. 4.

Fig. 6. It will be much potentiated if they are able to show the generation of unanchored K63 chain by Ube2N in in vitro ubiquitination assay.

Response to reviewer's remarks

Replies to Reviewer #1

We appreciate the reviewer's critical comments and would like to provide our point-to-point answers as following. Figures shown are labeled with R-Fig.

This manuscript reports two important findings that (1) Riplet, not Trim25, is the E3 ligase that activates RIG-I, (2) Riplet functions together with the two distinct E2 enzymes, Ube2D3 and Ube2N, to use both unanchored and anchored Ub chains for RIG-I activation. The authors used cell free assays to dissect the role of individual E3 ligases in RIG-I activation. Subsequent biochemical fractionation analysis led to the identification of Ube2D3 and Ube2N as the partner E2 enzymes that work with Riplet. These findings are novel and important, and are well supported by the data. However, there are a few areas that need further clarification and additional experiments to avoid confusions.

1. While the authors' findings are intriguing, they are also contradictory to some of the previous reports (by J.U. Jung, Michaela Gack and James Chen) suggesting the major role of Trim25 in RIG-I activation. The authors should address why this discrepancy might have occurred and what the authors think is the actual role of Trim25 in the RIG-I signaling pathway. Even within this manuscript, somewhat confusing/contradicting data have been presented - for example, Fig. 6c shows that Trim25 can replace Riplet in MAVS aggregation, at least in this in vitro assay.

Answer: The previous reports could be due to an off-target by siRNA oligoes. We tested the oligoes used in these studies and found that RIG-I activation was affected (R-Fig 1) and MAVS downstream signaling was affected as well

(R-**Fig 2 & 3**). Most importantly, reduction of interferon production by the oligoes could not be rescued by ectopically expressed TRIM25 (R-**Fig 4**). So we conclude previous reports might be due to an off-target effect.

R-**Fig 1**

R-**Fig 2**

R-**Fig 3**

R-**Fig 4**

In our *in vitro* assay, RIG-I can be activated by K63-linked polyubiquitin chains, which could be synthesized by those different ubiquitin E3 ligases, or even chemically synthesized chains. Therefore, in Fig.6c, RIG-I could be activated by chains synthesized by both Riplet and TRIM25.

2. Fig. 2c: the text does not match the legend. Data in Fig. 2c shows that you need both fractions A & B to stimulate MAVS aggregation. However, the text (in line 169-174 on page 9) suggests that the fraction A is sufficient. This is more confusing as it was followed by the statement that the fraction B contains E1, which is clearly necessary for RIG-I activation.

Answer: Both fractions A and B are required for RIG-I activation. Since fraction B contains E1, we substituted fraction B with recombinant E1 in the following purification procedure. We revised the description and are sorry for the confusion.

3. Fig 2c: Ube2N is thought to function as a complex with Ube2V1. Which fraction contains Ube2V1? Does the fraction requirement supports the notion that Ube2N needs Ube2V1?

Answer: Indeed, Ube2V1 is required for Ube2N to function properly. Fraction A from Fig2c contains Ube2V1 (**R-Fig 5**). Ube2V1 is also evidenced in our purified fraction subjected for mass spectrometric analysis (**R-Fig 6**. i.e. Supplementary Figure 2e).

R-Fig 5

R-Fig 6 i.e. Supplementary Figure 2e

4. In general, the relationships among the E2 enzymes in Fig 3 were not clear. There are apparent discrepancies among the results in Figs 3 & 4 on the role of Ube2D3, Ube2E1 and Ube2N. For example, Fig. 3b suggests that Ube2E1 is sufficient to activate MAVS, but Fig3d suggests that it is Ube2N & Ube2D3 that are important. If results in Fig 3b are due to the usage of enzymes at non-physiologically high concentration, is Fig 3b necessary? Another confusing part is the role of Ube2N. In Fig 3c, KO of Ube2N moderately reduced the IFN induction, but further deletion

of 2D1,2,4 & E1 restored the IFN induction activity. In Fig. 4a, KD of *Ube2N* alone was sufficient to abolish IFN induction. How can the authors reconcile all these results?

Answer: Using recombinant protein to verify the reconstituted assay in vitro is a standard procedure in biochemical purification study, so that Fig 3b is necessary. Our results showed that *Ube2D1/2/4* play a negative role in IFN production, so that *Ube2D1/2/4^{-/-}* & *Ube2N/E1^{-/-}* cells produced more IFN than *Ube2N^{-/-}* cells (R-Fig 7).

R-Fig 7

Ube2D1/2/4 may contribute to synthesise K48 linkage ubiquitin chain and mediate the degradation of essential factors in this pathway. Our data showed *Ube2N* alone is required in IFN induction in mouse cells (including MEF, PEM, BMDM), in contrast to that both *Ube2D3* and *Ube2N* are required for IFN production in human cells (293T), which is species-specific (R-Fig 8. i.e. Supplementary Figure 5).

R-Fig 8 i.e. Supplementary Figure 5

5. Were the binding assays in Fig. 5b and 5c done with or without virus infection (or with viral RNAs)? If not, the experiment should be repeated with viral RNAs (the condition where native Riplet-RIG-I interaction is expected to occur).

Answer: We repeated experiment shown in Fig.5b and 5c in the presence of VSV infection (**R-Fig 9**. i.e. Supplementary Figure 6a and 6b). Upon viral infection, the binding of RIG-I (full length) to Riplet was enhanced.

R-Fig 9 i.e. Supplementary Figure 6a and 6b

6. The authors used cell free assay results in Fig 6D & E to argue that Riplet makes K63-linked Ub chains. This should be also shown by cellular assays (for example by comparing RIG-I ubiquitination in Riplet-deficient and -sufficient cells with linkage specific antibody against Ub).

Answer: The study was performed as suggested and shown WB in **R-Fig 10** i.e. Supplementary Figure 6d.

R-Fig 10 i.e. Supplementary Figure 6d

7. The proposed roles of anchored and unanchored Ub chains seem similar to what was previously described in the report of the CARD-Ub complex structure (Peisley et al Nature 2014). The results in Fig 7 and the role of Ub should be discussed in the context of the structural mechanism of Ub mediated oligomerization of CARD.

Answer: We include the discussion “Both conjugated (to RIG-I) and unanchored polyubiquitin chains are capable of activating RIG-I, which is conceivable in light of that conjugated polyubiquitin chains could activate RIG-I *in trans*. As a matter of fact, this view is consistent with the structural study showing that conjugated polyubiquitin chains to RIG-I 2CARD is compatible with their non-covalent binding to 2CARD, which may promote the formation of 2CARD tetramer to assemble MAVS CARD filament (Peisley et al Nature 2014)”.

8. Fig 7e. In the absence of Ube2D3, Ub synthesis and RIG-I activation is expected to be mediated by Ube2N (unanchored Ub chains). K96 and K172 are the conjugation sites, and are not involved CARD-Ub binding.

However, Fig 7e shows that K63/172R RIG-I has no activity in Ube2D3 KO. What is the explanation of the requirement for K96 and K172 in a Ube2N-mediated activation of RIG-I?

Answer: We reasoned the ubiquitin conjugated sites K96/172 also play a role in unanchored Ub chains binding during RIG-I activation. To verify this hypothesis, we used flag-tagged RIG-I proteins (WT or point mutations) to incubate with ubiquitin reaction mixture stopped by NEM, then the RIG-I protein was purified by M2 beads and subjected to SDS-PAGE to test free ubiquitin chains-binding. When K96/172 were mutated, RIG-I nearly lost its ability to bind free Ub chains (**R-Fig 11**. i.e. Supplementary Figure 7a). So these lysine residues were not only conjugation sites of ubiquitin, but also key residues for free Ub chain binding.

R-Fig 11 i.e. Supplementary Figure 7a

Reviewer #2 (Remarks to the Author):

We appreciate the reviewer's critical comments and would like to provide our point-to-point answers as following.

To date, several studies have shown that ubiquitin-dependent processes play a critical role in coordinating the assembly of functional signaling complexes both upstream and downstream of MAVS. Upstream MAVS, K63-linked polyubiquitin chains have been shown to regulate RIG-I activation. TRIM25, RIPLET, TRIM4 and MEX3C are all E3 ligases that have been implicated in the K63-linked ubiquitination of RIG-I (the KD and/or KO of each of the above-mentioned E3 ligases has been shown to significantly reduce RIG-I-dependent IFN induction). In addition, in vitro cell-free experiments showed that unanchored K63-linked polyUb chains generated by TRIM25 are also able to activate RIG-I signaling. However, how different E2 and E3 enzymes work together to modulate signal transduction in different species and cell types, and how different stimuli can regulate the activity of such enzymes during infection still remain to be elucidated.

In this study, the authors take advantage of in vitro assays to dissect the early events of RIG-I activation. The authors show that in their cellular systems (293Ts and MEFs), RIPLET is the only E3 ligase that is required for MAVS aggregation as well as downstream signaling upon viral infection. By performing biochemical fractionation coupled with mass spec analysis, the authors then identify Ube2D3 and Ube2N as the two critical E2 enzymes required for Riplet-dependent RIG-I activation. Ube2N being essential for MAVS aggregation in MEFs, whereas Ube2D3 and Ube2N playing redundant roles in 293Ts. Based on data from cell-free assays, the authors then suggest that while the Riplet-Ube2N

pair preferentially catalyzes the synthesis of unanchored K63-linked polyUb chains, the Riplet-Ube2D3 pair promotes covalent conjugation of polyUb chains to RIG-I.

These findings are important to better understand the molecular mechanisms regulating RIG-I activation by Riplet. However, some of the data in this manuscript are not in agreement with the pre-existing literature, and more convincing data should be provided to support them. Additional experiments in more relevant cellular systems are needed to confirm the relevance of these findings.

Specific issues are detailed below:

1) In Figure 1d, Riplet CRISPR KO cells show a dramatic defect in MAVS aggregation. Did the authors check whether Riplet depletion affects the expression levels of RIG-I and other known critical proteins in the RIG-I activation cascade such as TRIM25, MAVS etc.?

Answer: We performed immunoblotting for other known critical proteins in the 293T and *Riplet*^{-/-} cell line and found that there was no effect on them (R-Fig 12. i.e. Supplementary Figure 1b).

R-Fig 12 i.e. Supplementary Figure 1b

2) In Figure 1e, TBK1 KO cells should be included as control. A WB with the expression levels of MAVS should also be included.

Answer: We repeated the experiment performed in figure 1e and included *Tbk1^{-/-}* cell as a control (R-Fig 13. i.e. Figure 1e). In addition, WB showed the protein level of MAVS (R-Fig 14. i.e. Supplementary Figure 1e).

R-Fig 13 i.e. Figure 1e

R-Fig 14 i.e. Supplementary Figure 1e

3) In Figure 1f, the authors utilize CRISPR KO MEFS that were not previously characterized. In contrast with what has been observed by several other groups, TRIM25 depletion did not affect IFN induction upon viral infection. Can the authors provide WB for these cell lines and comment on these differences? Were MEFS reconstituted with human or mouse TRIM25/RIPLET? Please specify.

Answer: WB for these cell lines are shown in (R-Fig 15. i.e. Supplementary Figure 1g). MEF were reconstituted with human TRIM25 or Riplet in our study.

R-Fig 15 i.e. Supplementary Figure 1g

4) A WB for the DKO cells used in figure 2B should also be provided.

Answer: The WB of DKO cells was showed in (R-Fig 16. i.e. Supplementary Figure 2b).

R-Fig 16 i.e. Supplementary Figure 2b

5) In Figure 3c, *Ube2N* KO cells exhibited a better phenotype than *Ube2Ds,N&E1* KO cells. Can the authors comment on this? Also, the differences in IFN induction in *Ube2N* KO cells do not correlate with the differences in MAVS aggregation shown in figure 3d.

Answer: Our results showed that *Ube2D1/2/4* play a negative role in IFN production, so that *Ube2D1/2/4^{-/-}&Ube2N/E1^{-/-}* cells produced more IFN than *Ube2N^{-/-}* cells (R-Fig 17).

R-Fig 17

In the *Ube2N^{-/-}* cells, MAVS aggregation was similar to WT cells (Figure 3d), this data suggested that *Ube2N* and *Ube2D3* play a redundant role in RIG-I activation. In addition, plenty work have shown that *Ube2N* plays a very

important role in MAVS downstream signaling. Indeed, when MAVS was overexpressed in WT and *Ube2N*^{-/-} cells, we found that the IFN-β induction in the knockout cell was lower than WT cell, suggested MAVS downstream signaling was crippled by Ube2N deficiency (R-Fig 18).

R-Fig 18

6) Are the differences observed in the Ube2N and Ube2D3 requirement between 293T and MEFs species-specific or cell type specific?

Answer: We further determined the requirement of Ube2N and Ube2Ds in RIG-I signaling as suggested in mouse primary cell lines, including peritoneal macrophage and bone marrow-derived macrophage, which is consistent with and confirm our results from MEFs (R-Fig 19. i.e. Supplementary Figure 5a). So the requirement of various E2s in RIG-I activation may be species-specific.

R-Fig 19 i.e. Supplementary Figure 5a

7) A role for Ube2D3 (Ubc5c) and Ube2N (Ubc13) in the RIG-I-dependent MAVS activation was previously shown by Zeng et al., 2009; Zeng et al.

2010; and Sanchez et al., 2016. This information should be included in the manuscript and discussed.

Answer: We include the discussion “Our findings are consistent with previous reports showing that Ube2D3 and Ube2N are critical for RIG-I signaling in siRNA-mediated knock-down experiments^{25,37}”.

8) In figure 5, the authors address the interaction between Riplet, RIG-I, Ube2D3 and Ube2N by co-ip experiments in 293T cells and show that the 2CARD domain of RIG-I and the SPRY domain of RIPLET are required for their interaction. Similar co-ips studies were previously performed (Gao et al., 2009; Oshiumi et al., 2009; Oshiumi et al., 2013) and different domains appeared to be critical for RIG-I/RIPLET interaction (Oshiumi et al., 2009; Oshiumi et al., 2013). The authors should comment on the differences observed in these studies. A RIPLET w/o SPRY domain control should be included in panel 5d.

Answer: Some of previous reports are contradictory to each other, which might be due to different experimental conditions and might be beyond our scope to address the discrepancy. In our study, Riplet w/o SPRY domain control was now included in Figure 5d (**R-Fig 20**).

R-Fig 20 i.e. Figure 5d

9) *The critical role of RIPLET, Ube2D3 and Ube2N in regulating RIG-I activation should be studied in more relevant cell types such as macrophages and DCs that are the main source of Type I IFNs in vivo. The role of TRIM25 in primary cells should also be determined. Riplet ko (Oshiumi 2010) and Ube2N conditional Ko mice (Yamamoto et al., 2006) have been previously described. Primary cell lines from these mice could be used to address this issue. Figure 5 should be also be complemented with endogenous co-ip studies in more relevant cells.*

Answer: We took knockdown approaches to address these questions, as knockout mice are currently not available to us. To determine the involvement of Ube2N and Ube2D3 in the primary cells, we knockdown these genes expression in mouse primary cell lines isolated from C57 mouse (**R-Fig 21**. i.e. Supplementary Figure 5a.). The result indicated the requirement of Ube2N in RIG-I signaling in PEM and BMDM cells, which is consistent with and confirm our results from MEFs. We also verified the requirement of Riplet but not TRIM25 in MEF, PEM, BMDM cells (**R-Fig 22**. i.e. Supplementary Figure 5c & e).

R-Fig 21 i.e. Supplementary Figure 5a

R-Fig 22 i.e. Supplementary Figure 5c & e

Our antibodies did not work for endogenous co-IP, so following or future work might provide more insight into this issue on endogenous molecules when proper reagents are available.

10) Figure 7 does not convincingly prove that residues 48/96/172 are indeed covalently ubiquitinated by the riplet-ube2d3 pair in cells upon VSV infection.

Answer: We provided more data to show that RIG-I is ubiquitinated by Riplet in response to viral infection (**R-Fig 23**, i.e. Supplementary Figure 6d). Our mass spectrometric result revealed the ubiquitination sites of K48/96/172, and mutations in three sites abolished RIG-I ubiquitination (Figure 7b) and severely crippled RIG-I signaling. Mutations in any one of these three sites affected RIG-I activity to different extent (**R-Fig 24**). Therefore, we believe that residues 48/96/172 are critical for the Riplet-Ube2D3 pair to activate RIG-I in cells.

R-Fig 23 i.e. Supplementary Figure 6d

R-Fig 24

Reviewer #3 (Remarks to the Author):

We appreciate the reviewer's critical comments and would like to provide our point-to-point answers as following.

RIG-I-like receptors bind viral RNA and initiate the antiviral immune response through their interaction with mitochondrial antiviral signaling protein MAVS. MAVS on the mitochondrial membrane forms prion-like aggregates that robustly activate downstream kinases and transcription factors. RIG-I activation can be regulated by the K63-linked polyubiquitination mediated by TRIM25 and Riplet E3 ligases. In addition, a critical role of unanchored K63 chains on the RIG-I activation was also shown. In this study the authors utilized a cell-free assay for viral RNA-triggered activation of RIG-I and MAVS aggregates and demonstrated that Riplet is the only E3 ligase that is required for activation of RIG-I to form MAVS aggregates. They also showed that Riplet-Ube2D3 (Ubc5c) promotes covalent conjugation of K63 chains to RIG-I while Ube2N (Ubc13) rather generates unanchored K63 polyubiquitin chains. Overall, biochemical data using a cell-free assay are convincing showing a nice correlation between MAVS aggregates and dimerization of IRF-3. However, new additional mechanistic information seems to be limited. Comments are below.

Fig. 1f. Reduction of IFN- β production was not observed in trim25 knockout cells, which is inconsistent to the previous findings (Gack et al., 2007). Verify these findings by reconstituting Flag-TRIM25 into the trim25 knockout cells. The authors may also try trim25 shRNA or siRNA.

Answer: We have reconstituted Flag-TRIM25 into Trim25^{-/-} cells (Figure 1f), and the result showed no effect on the anti-viral pathway. To further address

the question, we use si-RNA mediated TRIM25 knockdown in 293T cells, which were used in previous reports (Gack et al., 2007). We found that RIG-I activation was affected (R-Fig 25) and MAVS downstream signaling was affected as well (R-Fig 26&27). Most importantly, reduction of interferon production by the oligoes could not be rescued by ectopically expressed TRIM25 (R-Fig 28). Therefore, we conclude previous reports might be due to an off-target effect.

R-Fig 25

R-Fig 26

R-Fig 27

R-Fig 28

Fig. 3. All of Ube2Ds, Ube2D1-D4, are able to activate RIG-I generating MAVS aggregates in 3a. In addition, three different concentrations of E2 are almost equally capable of activating MAVS and IRF-3. Although the authors showed that dual disruption of Ube2Ds and Ube2N abrogated the IFN induction (3c) and MAVS aggregation (3f), authors should show whether ectopic expression of Ube2D1, 2, 4 into the Ube2D1,2,4-/-&D3

siRNA cells did not restore the IFN induction and MAVS aggregates to verify the specific role of Ube2D3 in RIG-I activation. In 3c, IFN production in UbeD1,2,4&N, E1-/- cells was twice higher than those of UbeD1,2,4 -/-. Why? Is it reproducible?

Answer: We performed Ube2D1,2,3,4 ectopic expression in the *Ube2D1,2,4,N^{-/-}* & D3 siRNA cell line (**R-Fig 29,30,31**). Ube2D1,2,4 overexpression in the cells slightly induced the expression level of IFN-β in the presence of viral infection, which was still much lower than D3 did. We think the slight induction by Ube2D1,2,4 shown in this assay may due to the ectopic expression or non-specific effect.

R-Fig 29

R-Fig 30

R-Fig 31

We repeated the experiment in Figure 3c, it is reproducible. Our results showed that Ube2D1/2/4 play a negative role in IFN production, so that *Ube2D1/2/4*^{-/-}&*Ube2N/E1*^{-/-} cells produced more IFN than *Ube2N*^{-/-} cells (R-Fig 32).

R-Fig 32

Fig. 4. In a cell-free assay, Ube2Ds and Ube2N play redundant roles in MAVS aggregation. In Fig. 4, authors showed that Ube2N only play an essential role for RIG-I mediated MAVS activation. It is confusing. The authors should describe better differences in systems. In addition, please clarify whether sh-Ube2D 3 or sh-Ube2D1/2/3 is applied to Fig. 4.

Answer: Our data showed Ube2N alone is required in IFN induction in multiple mouse primary cells (including MEF, PEM, BMDM), in contrast to that both

Ube2D3 and Ube2N are required for IFN production in human cells (293T). Therefore, we conclude that the result is species-specific (R-Fig 33 i.e. Supplementary Figure 5).

R-Fig 33 i.e. Supplementary Figure 5

In Figure 4 we used si-RNA to knockdown all Ube2Ds, in which the Ube2D2,3,4 shared the same targeting sequence, and Ube2D1 has its unique targeting sequence.

Fig. 6. It will be much potentiated if they are able to show the generation of unanchored K63 chain by Ube2N in in vitro ubiquitination assay.

Answer: We performed the experiment suggested and the data was shown as below (R-Fig 34. i.e. Supplementary Figure 6c)

R-Fig 34 i.e. Supplementary Figure 6c

Response to Reviewers:

Reviewer #1 (Remarks to the Author):

Overall, the authors have made a significant improvement to strengthen the paper. The revised manuscript is appropriate for publication in Nature Communication.

Reviewer #2 (Remarks to the Author):

The authors have performed additional experiments and taken care of the comments of this reviewer

Reviewer #3 (Remarks to the Author):

The authors fully addressed the comments raised by the reviewer.